



# Calibration and Validation of the Polarimetric Radio Occultation and Heavy Precipitation experiment Aboard the PAZ Satellite

Ramon Padullés[1], Chi O. Ao[1], F. Joseph Turk[1], Manuel de la Torre Juárez[1], Byron Iijima[1], Kuo Nung Wang[2,1], and Estel Cardellach[3]

[1]Jet Propulsion Laboratory, California Institute of Technology, Pasadena, CA, 91109
[2]Scripps Institution of Oceanography, La Jolla, San Diego, CA
[3]Institut de Ci'encies de l'Espai, Consejo Superior de Investigaciones Científicas, Institut d'Estudis Espacials de Catalunya

**Correspondence:** ramon.padulles.rullo@jpl.nasa.gov

**Abstract.**

This manuscript presents the calibration and validation studies for the Radio Occultations and Heavy Precipitation experiment aboard the PAZ satellite. These studies, necessary to assess and characterize the noise level and robustness of the $\Delta\Phi$ observable of Polarimetric Radio Occultations (PRO), confirm the good performance of the experiment and the capability of this technique in sensing precipitation. It is shown how all the predicted effects that could have an impact into the PRO observables (e.g. effect of metallic structures nearby the antenna, the Faraday Rotation at the ionosphere, signal impurities in the transmission, altered cross polarization isolation, etc.) are effectively calibrated and corrected, and they have a negligible effect into the final observable. The on-orbit calibration, performed using an extensive dataset of free-of-rain and low ionospheric activity observations, is successfully used to correct all the collected observations, which are further validated against independent precipitation observations confirming the sensitivity of the observables to the presence of hydrometeors. The validation results also show how vertically averaged $\Delta\Phi$ can be used as a proxy for precipitation.

## 1 Introduction

The Radio Occultations and Heavy Precipitation (ROHP) experiment on board the PAZ satellite was switched on on May 10th, 2018, after a successful launch on February 22nd, 2018. For the first time, Radio Occultations (RO) are acquired at two linear polarizations with the aim to detect heavy precipitation. The technique, called Polarimetric RO (PRO), consists in measuring the phase difference between the horizontal (H) and vertical (V) components of the electromagnetic field coming from the Global Positioning System (GPS) satellites in occulting geometry (Cardellach et al., 2014). This is an augmentation of the capabilities of the well known RO technique (e.g. Kursinski et al., 1997; Hajj et al., 2002). The first preliminary results obtained during the first five months of data confirm that the measurement is sensitive to precipitation (Cardellach et al., 2019).



H and V components are measured independently, yet synchronously, with a dual linear polarized antenna pointing towards the limb of the Earth in the satellite's anti-velocity direction. The rays, curved and delayed as they penetrate into deeper layers of the atmosphere (with higher density), reach the receiver in occultation geometry. The delay of the rays can be precisely tracked, and information about the thermodynamic state of the atmosphere can be retrieved (e.g. vertical profiles of temperature,

pressure and water vapor pressure) as in standard RO. The fact that in the PAZ satellite the incoming electromagnetic field is acquired at two linear and orthogonal polarizations allow us to retrieve information about media that introduce a differential phase shift between the horizontal and vertical components of the propagating electromagnetic waves. The media introducing this effect are mainly hydrometeors that flatten due to air drag as they fall. The scattering of electromagnetic waves by these asymmetric hydrometeors introduces a differential change in phase between the H and V components, that is proportional

to the amount and size of the hydrometeors. Therefore, this experiment represents the first technique able to retrieve vertical information of precipitation and the thermodynamic state of the surrounding area within the same measurement, from space.

The first analysis (Cardellach et al., 2019), was focused only on the ability of the technique to detect hydrometeors, and a thorough calibration of the receiving system is required before more quantitative results can be obtained. The calibration of the receiving system is critical in assessing the uncertainty level of the measurement, and therefore to associate geophysical

quantities like rain intensity to each phase measurement (Cardellach et al., 2017). The purpose of the calibration of the receiving system is to remove the systematic effects unrelated to hydrometeors (Tomas et al., 2018). These include the ionospheric effect into the polarimetric signal, the impurity of the transmitted signal, the ambiguity introduced by the receiver tracking two independent signals, and any other instrumental effects. In addition, the environment around the receiving antenna needs to be characterized. Before launch, a metallic structure had to be added to the satellite in order to adapt it to the a new launch vehicle

(see Fig. 1 - top). This structure sits 30 mm above the antenna, and covers part of the field of view. It introduces a systematic effect that depends on the angle of arrival of the signal at the antenna, and changes the antenna patterns from those measured in an anechoic chamber before installation (Cardellach et al., 2014). It is also very likely that the metallic structure has worsened the cross polarization isolation of the antenna.

In order to calibrate the signal, all the available data from May 10th 2018 to March 30th 2019 are accumulated and grouped

based on the corresponding precipitation information: the data are classified into clear skies and cloudy-rainy scenes. This classification is performed using information from the National Centers for Environmental Prediction / Climate Prediction Center (NCEP/CPC) Infrared Brightness temperature (Janowiak et al., 2017) and Integrated Multi-satellitE Retrievals for the Global Precipitation Mission (GPM IMERG) (Huffman, 2017) rain rates. This allows us to determine the uncertainty of the measurement when no hydrometeors are present, so that there is nothing expected to introduce changes in the differential phase

between H and V. Then, the effect of the ionosphere (through Faraday rotation) is assessed, using co-locations between the simulated RO ray-paths and the International Geomagnetic Reference Field (IGRF) Earth's magnetic field model (Thébault et al., 2015) and the International Reference Ionosphere (IRI) climatology for the electron density (Bilitza et al., 2017). Finally, the uncertainty and biases introduced by the antenna are characterized, so that they can be corrected in each measurement.





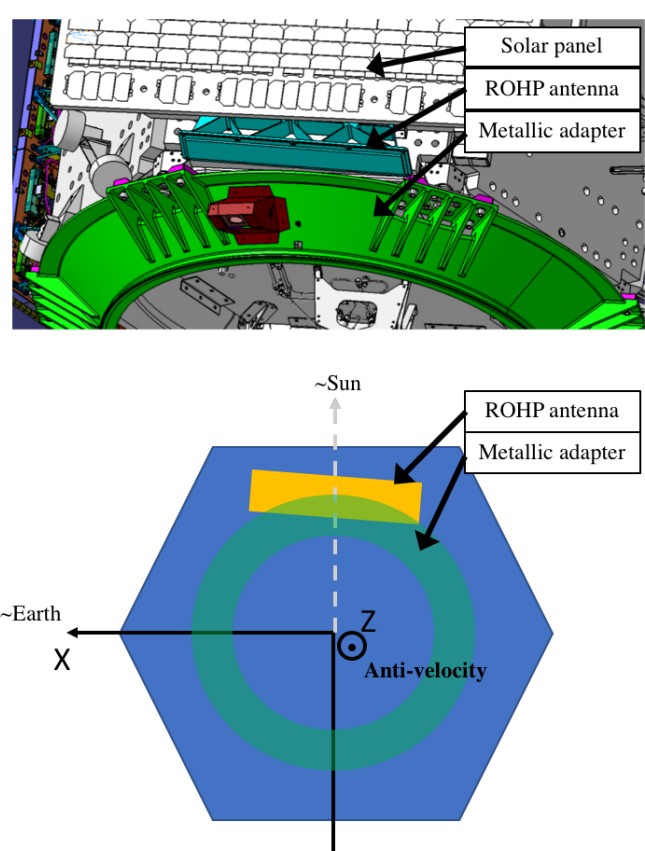

**Figure 1.** Top: Schematic draw of the metallic adapter structure (green) over the ROHP antenna (blue), and their position in the satellite. Image provided by Hisdesat. Bottom: Sketch of the satellite body (blue), ROHP antenna (yellow) and the metallic adapter (green), and the body fixed cartesian reference frame with the X axis pointing toward the direction of the Earth, the Z axis pointing towards the anti-velocity direction, and Y axis being the third orthogonal component to define the reference frame.

Once the receiving system has been calibrated and the data accordingly corrected, these new observables are validated using the GPM products mentioned above. The results of the validation are compared with what was obtained in Cardellach et al. (2019) and also with the predicted performance from the simulations in Cardellach et al. (2014).

## 2 Data

5 The data collected by the PAZ satellite are down-linked by Hisdesat. The Institut de Ciències de l'Espai (ICE), Consejo Superior de Investigaciones Científicas, Institut d'Estudis Espacials de Catalunya (CSIC, IEEC) collects, owns the data and provides access to the servers at the Jet Propulsion Laboratory (JPL). At the JPL, the raw data are processed and converted to level 1 RO products which are finally analyzed.



## 2.1 Polarimetric Phase Calibration

The JPL designed IGOR+ receiver installed in PAZ collects RO data at a rate of 50Hz. Each RO is tracked independently in the two ports dedicated to the H and V polarized antennas. Therefore, each port output is processed independently. Raw phase data can be converted into excess phase ($\phi$) using Precise Orbit Determination (POD): the excess phase identifies the phase

delay of the incoming electromagnetic field after removing the geometric contribution (i.e. the distance between satellites and their relative movement) (Hajj et al., 2002). Errors due to satellite and receiver clocks are also corrected. Hence, $\phi$ is due to atmospheric effects. Its variation as a function of time, i.e. Doppler shift, is the main observable for ROs.

The signal amplitude, or SNR, and $\phi(t)$ from each antenna port are used to obtain the bending angle as a function of the impact parameter ($\alpha(a)$) using the Canonical Transform method (Gorbunov, 2002). Then, under the assumption of an

spherical symmetric atmosphere, the inverse Abel transform (Fjeldbo et al., 1971) is used to retrieve the profile of refractivity as a function of geometric height ($N(h)$). This process is the same one applied to in conventional GNSS RO, and it can also be applied in this case.

The main new observable for PRO the difference between $\phi(t)$ of both ports is:

$$\Delta\Phi = \phi_{\mathrm{H}} - \phi_{\mathrm{V}}. \tag{1}$$

$\Delta\Phi(t)$ should be constant in time if nothing along the ray-path that is introducing a differential phase shift. Notice that the absolute phase difference between the H and V components of a RHCP electromagnetic wave should be $\pi/2$, however, since the two components are tracked independently and what remains is the excess phase, this difference is no longer $\pi/2$, but a constant random number. When precipitation is present in any point along the raypath, $\Delta\Phi(t)$ should increase.

Even though the initial processing of the raw data corrects for cycle slips (i.e. changes in $\phi$ of more than one cycle in

consecutive measurements), after computing $\Delta\Phi(t)$ some jumps in the observable are detected. These jumps are associated to cycle slips that remained uncorrected before, or appeared after computing the difference between the two $\phi(t)$ (h and v). Therefore, the $\Delta\Phi(t)$ is also corrected for cycle slips in the following way:

$$\Delta\Phi(t) = \arctan(\tan(\Delta\Phi(t))) \tag{2}$$

This approach should correct the cycle slips remaining in the data. However, this approach can still introduce a $2\pi$ jump in

the phase if this is too close to $\pm\pi/2$, although this is an infrequent situation. Since the $\Delta\Phi$ tends to follow a rather smooth variation in the presence of precipitation, events inducing such large $\Delta\Phi$ values can be easily identified and treated accordingly.

For each port, data are processed to obtain $N(h)$. To assign a height to each time measurement (e.g. excess phase or SNR) is complicated, specially when atmospheric multipath is present. To do so we rely on the inverse Abel transform and we assign a tangent height (the height of the tangent point of each ray) to each phase and SNR measurement, $\Delta\Phi(h_{\mathrm{t}})$ and $\mathrm{SNR}(h_{\mathrm{t}})$. As

a convention, the height that is assigned to each time is the mean of the heights obtained in the H and V ports at that time. To set a common reference for all the data that is independent on the initial phase of the receiver, we set the zero at 30 km, and





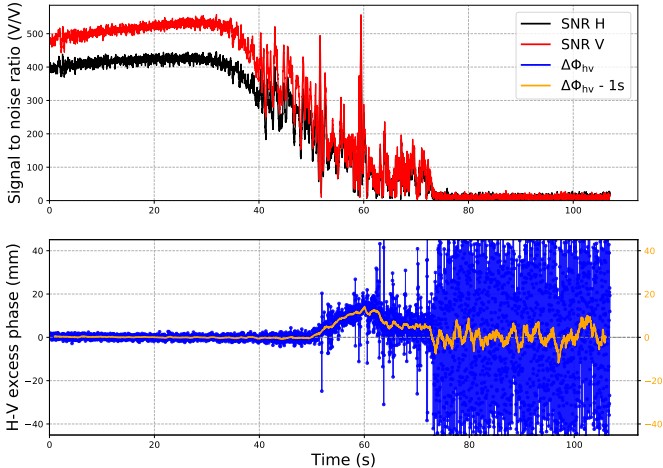

**Figure 2.** Example of one Polarimetric RO observation, corresponding to id 20181024_1403_paz_gps68. The RO tangent point is located at (30.6S, 102.2W). (Top) SNR for H (black) and V (red) ports as a function of time. (Bottom) Raw differential phase shift between the H and V excess phase observables (blue) and the corresponding 1 second smoothed measurement (simple average running window). After T=72s, the SNR is too low to keep track of the phase.

therefore $\Delta\Phi = \Delta\Phi - \Delta\Phi(h_\mathrm{t} = 30)$. At this height we know that there is no rain, clouds or ice that could infer any measurable differential phase shift. Therefore, all measurements are relative to that height.

The whole processing is applied to 59,704 occultations, of which a total of 42,209 pass through the JPL quality control. The quality control is passed if the retrieved refractivity profiles between 0 and 30 km (for both H and V) are within 10% of the
5  co-located NCEP Global Forecast System (GFS). Those that do not pass the quality control are discarded.

## 2.2 Ray tracing

In order to identify the region that is being sensed by the PRO, we need to define the RO plane. The RO plane is formed by all the rays from the GPS transmitter to the receiver. This plane is slant rather than vertical due to the relative movement between the GPS and the LEO, which are not coplanar. To define a realistic RO observation plane, we account for realistic rays between
10  the GPS and the LEO obtained using a ray tracing software that provides the ray's trajectories for every time step of the PRO event. The ray tracing uses the actual retrieved refractivity profile to account for the bending of the rays.

The whole set of trajectories, e.g. (time, lon, lat, height), can be used to identify the regions traversed by the rays, and therefore perform realistic and accurate co-locations between different datasets (like precipitation) for reliable calibration and validation of the experiment.



## 2.3 Co-location of PRO observations with GPM constellation products

For the calibration and validation part of the experiment, the co-locations with precipitation products is crucial. It provides an independent measure on whether an observation might have been affected by rain or not. Since the effect of rain is the objective and it should exhibit a clear distinct signature from the no rain events, the calibration of the receiving system should be done with the rain free events. Therefore, the co-locations with precipitation products has to be as accurate as possible.

We consider that the IMERG precipitation products are the best suited for such co-location. First of all, these products provide information of precipitation covering between $\pm$ 60 degrees in latitude and all longitudes with a high spatial resolution (i.e. 0.1 $\times$ 0.1 deg), so offering the best global coverage among precipitation products. The 30-minute time resolution of IMERG products is also acceptable for the co-locations that we need.

For the first analysis of the ROHP-PAZ data (Cardellach et al., 2019), where we aimed for a quick look of the sensitivity of PRO observations to precipitation events, we linked every occultation with the intensity of precipitation in the surrounding areas, using cells of fixed size and circularly shaped (e.g. 2 and 0.6 deg of diameter). Although this approach was effective, here we perform a more accurate co-location using the actual shape and orientation of the PRO sensed region. The region that is sensed by PRO observations can be approximated by a slant vertical plane (RO observation plane, see Sect. 2.2). This results in a sensed region that is long in the direction parallel to the line between the GPS and the LEO, but that also has a certain width in the cross direction when projected to the ground. Since the IMERG precipitation product only provides surface precipitation (2D), the ground projections of the RO observation plane is what we use to define the region in which we average the precipitation intensity. This region, however, is defined using only the portion of the RO observation plane below a certain altitude, since we only expect precipitation to have an effect to the lowest portions of the rays.

In order to use the projection to the ground of the RO observation plane, we need to assume that precipitation has some vertical structure, and the rays above the surface level can be affected by precipitation as well. Therefore, we use two different heights to define the maximum height at which the rays might be affected by the precipitation: 6 km and 12 km. These two altitude values define the portion of the RO plane that we project on the surface, and therefore the area in which we average precipitation. For example, when using the 6 km threshold, the RO plane to be projected on the surface will be defined only by the portions of the rays below 6 km. The higher the altitude, the larger the area will become. An example of the co-location strategy is shown in Fig. 3. Using the strategy described above we can reduce the cases where the ray does not cross precipitation, but it would be labeled as rainy in a 2° circular co-location, and reduce the uncertainty of the co-locations that might have appeared in Cardellach et al. (2019).

For the rest of the paper, when we refer to the precipitation associated to a PRO event, we will be referring to the average of the precipitation rain rate provided by IMERG inside the region sensed by that event (region defined by the red line in Fig. 3). Therefore, we do not make any distinction between the actual intensity and extension of the precipitation.

The same approach is used to obtain the information about the brightness temperature ($Tb$). We evaluate the $Tb$ provided by the NCEP/CPC Infrared products in order to add valuable information to the co-locations, since from $Tb$ we can determine the cloud top temperature (and approximated derived height) and have an idea of the development status of a precipitation




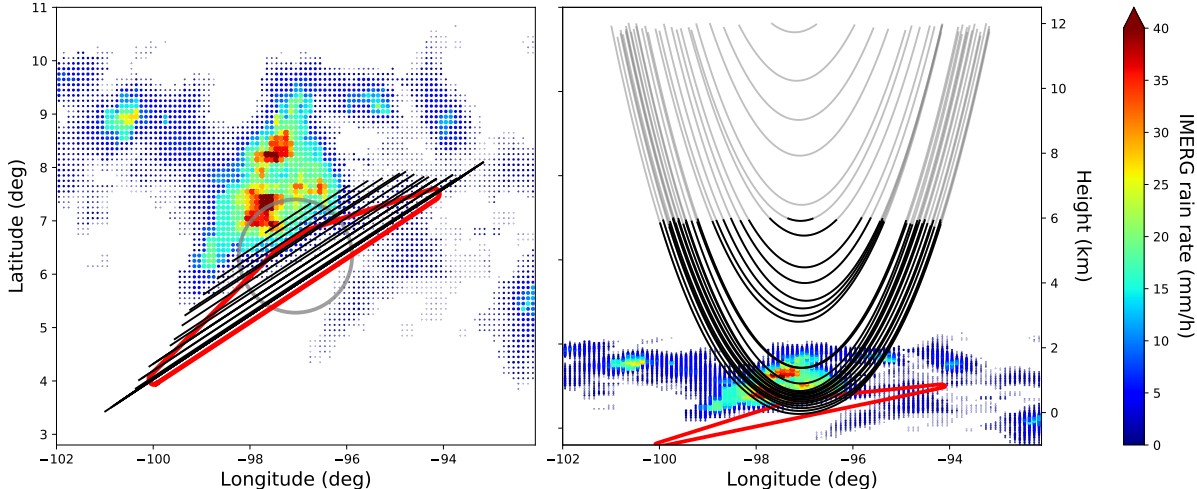

**Figure 3.** Example of the co-location strategy. Left: The black lines show the projection on the surface of the portion of the RO rays below an altitude of 6 km . As the observation descends, more portion of the rays happen to be below the set altitude thresholds, therefore the shortest segments represent the higher rays. The red line defines the region where the precipitation is evaluated. For comparison purposes, a circle (gray) of 2 deg of diameter - used in Cardellach et al. (2019) - is shown. The background color is the surface precipitation rain rates from IMERG at a $0.1 \times 0.1$ deg resolution. Right: The RO rays shown in a longitude - height projection. The darkest part of the rays represent the portion below 6 km. The background IMERG surface precipitation (same as in left panel) is shown here as a 3D projection, where x axis corresponds to longitude and y axis to latitude (all values are contained in the longitude - latitude plane). Only a few RO rays are shown here for illustration purposes.

structure. For this purpose, instead of retrieving the mean $Tb$ in the PRO sensed region, we collect the minimum $Tb$, more indicative of the cloud top properties of the tallest structure in the region.

This approach limits the number of occultations with precipitation information to those that reach below 6 km, and those located within $\pm 60$ deg of latitude (area covered by IMERG). Therefore, the total number of occultations co-located with precipitation is 16,292.

### 2.4 Co-location with IRI and IGRF

The ionosphere can have an effect into the differential phase shift observable (Tomas et al., 2018), through Faraday Rotation. It depends on the magnetic field and the electron density, so we need to know these quantities at any given point of the ray trajectories. Therefore, we co-locate the realistic (t, lon, lat, height) points (see Sect. 2.2) with the IRI for the electron density, and with the IGRF (Thébault et al., 2015) for the Earth's magnetic field. Knowing this information, we can compute the estimated Faraday Rotation that a given ray undergoes and estimate its effect into the $\Delta\Phi$ (as detailed in Sect. 4).



## 3 Antenna pattern

The antenna pattern characterizes the response of the antenna depending on the direction at which radio waves from GPS satellites arrive to the LEO. By having a good characterization, we can set the zero level of the measurements, i.e. the measurement obtained without anything affecting the signal. For this reason, we establish the on-orbit antenna pattern using only data that we know for sure that have not crossed precipitation, and were obtained under low ionospheric activity. In fact, this is not an actual antenna pattern, but it also contains some features arising from the transmitted/propagated signal effects. Therefore, it represents an "effective" antenna pattern.

The direction of arrival is given by the azimuth and elevation defined on a particular reference frame. To define such a reference frames we need to know, very precisely, the positions of the GPS that emits the radio wave, and the position and relative orientation of the PAZ antenna with respect to the emitter. To account for the relative orientation, we use the quaternions, provided along with the orbits data.

### 3.1 Definition of the reference frames

The three axis that define the reference frames are fixed in the body of the satellite. To account for the satellite orientation and maneuvers we use the quaternions, that precisely define the orientation of the satellite with respect to the center of the Earth Centered reference frame at any given time. The PAZ satellite orbits the Earth in a sun-synchronous orbit, with an inclination of $98°$. This means that the satellite has always a side facing the Sun. The principal instrument on PAZ, the Synthetic Aperture Radar (SAR), faces the Earth's surface and the PRO antenna is placed in the rear end of the satellite, facing the anti-velocity vector of PAZ. This configuration is depicted in Fig. 1. and is used to define the three principal axes of the reference frame:

- The Z axis is perpendicular to the antenna and therefore defines the normal vector to the antenna surface. In general, the Z axis points towards the same direction as the anti-velocity vector of the satellite (i.e. $-v_{\mathbf{sat}}$), but due to satellite maneuvers, the angle between Z and $-v_{\mathbf{sat}}$ can be as large as $4°$.

- The X axis points approximately towards the center of the Earth. However, this is not completely true due to the non-sphericity of the Earth, the non-circularity of the satellite's orbit and the satellite's maneuvers.

- The Y axis is defined to be perpendicular to both X and Z, and points approximately towards the opposite direction of the Sun, taking into account the aforementioned circumstances.

Once we have defined the three axes that form the reference frame, we can describe the GPS position in this reference frame: $\mathrm{gps} = (x_g, y_g, z_g)$.

### 3.1.1 The Antenna reference frame

The so defined Antenna reference frame is used for the antenna characterization. This reference frame is constructed on the XYZ axes defined above and is a spherical coordinate system. Therefore, is specified by a radial distance, a polar angle (or





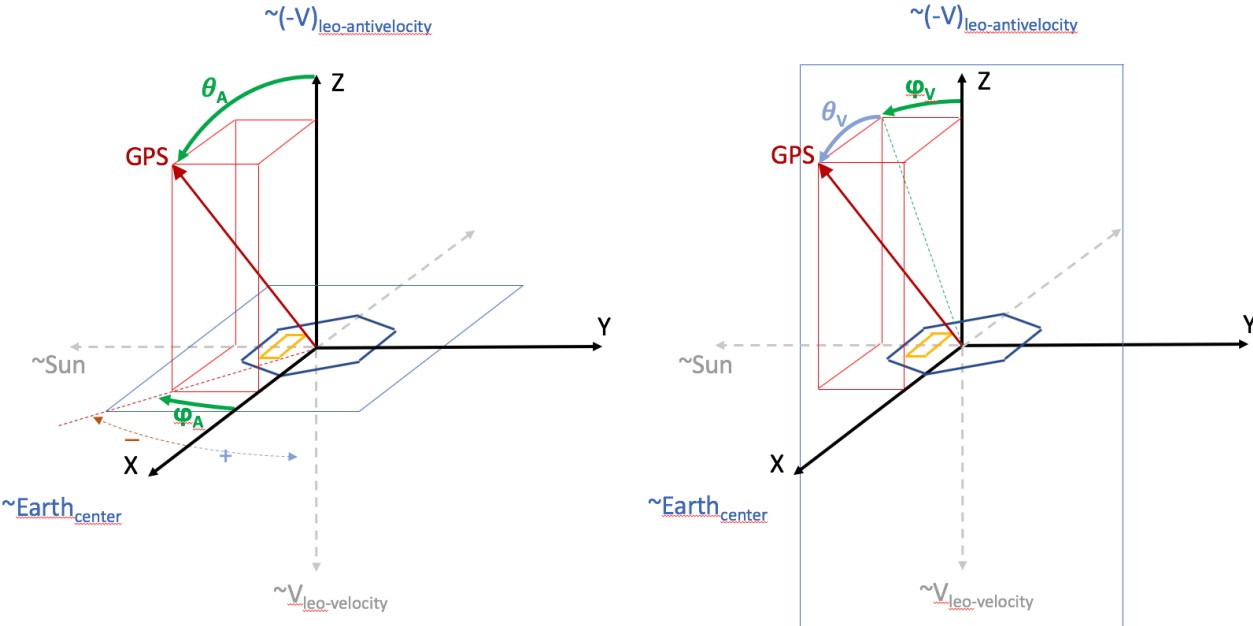

**Figure 4.** Graphical sketch of the reference frames defined in the text. The same XYZ axes (fixed in the satellite body frame) are used to define both reference frames. (Left) Antenna Reference frame: the principal plane where the location of the GPS satellite is evaluated is the XY, emphasized by a thin blue line; (Right) Velocity Reference frame: the principal plane where the location of the GPS satellite is evaluated is the YZ, emphasized by a thin blue line. The approximate directions at which the different axes point to are specified for reference for the reader.

inclination), and an azimuthal angle. The radial distance corresponds to the distance between PAZ satellite and the tracked GPS satellite. The polar angle, or inclination ($\theta_A$), corresponds to the angle between the Z axis and the origin - GPS vector. The azimuth angle ($\varphi_A$) corresponds to the projection of the inclination angle on the XY plane, containing the origin and orthogonal to the zenith. The positive values of $\varphi_A$ are defined such as the angle increases towards the positive Y. The formal

5   definition of the angles is as follows:

$$\varphi_A = \arctan\left(\frac{y_g}{x_g}\right) \tag{3}$$

$$\theta_A = \arccos\left(\frac{z_g}{\sqrt{x_g^2 + y_g^2 + z_g^2}}\right) \tag{4}$$

This reference frame is sketched in Fig. 4 - left.

### 3.1.2 The Velocity reference frame

10   It is also worth defining another reference frame, named here as the Velocity reference frame, used in the RO community and used to define the parameters set in the RO receiver aboard PAZ. Differently from the Antenna reference frame, whose





reference plane is the plane containing the antenna, here the reference plane is the YZ one, which is parallel to antenna's normal vector and (pseudo)-tangential to the Earth's surface. Once the reference frame is defined, the azimuth ($\varphi_V$) and elevation ($\theta_V$) angles can be defined as:

$$\varphi_V = \quad \arctan\left(\frac{y_g}{z_g}\right) \tag{5}$$

$$\theta_V = \quad \arctan\left(\frac{x_g}{\sqrt{y_g^2 + z_g^2}}\right) \tag{6}$$

This reference frame is sketched in Fig. 4 - right.

## 3.2 Antenna pattern characterization

To characterize the response of the antenna depending on the angle of incidence of the incoming radio waves, we use the $\varphi_A$ and $\theta_A$ based on the relative positions of the GPS and the LEO, without taking into account the bending angle of the ray by the atmospheric refractive index gradients. The consequence is that $\theta_A$ will be overestimated due to the fact that the actual rays bend and arrive to the antenna as if they were coming from the limb of the Earth while the actual GPS position is below the Earth's surface. Since we only use the positions, i.e. straight rays, the $\theta_A$ spans further down than it really is.

First of all, we look at the effective antenna pattern of the SNR for both the H and V antennas (Fig. 5). The SNR antenna patterns show a different behavior in the H and V antennas. Based on the measurements made in an anechoic chamber before the installation of the antenna (shown in Cardellach et al. (2014), Fig. 5 - top and center panels), the H antenna should perform slightly better than the V one, and have a maximum gain centered at $\varphi = 0$ decreasing towards the edges. However, we can see in Fig. 5 how the installation of the metallic structure changed this pattern. Now, the best performance is achieved by the V antenna, although the maximum gain is centered around $\varphi_A = -15\,\mathrm{deg}$. The lower performance at $\varphi_A > +20\,\mathrm{deg}$ is most likely due to the blockage by the metallic structure. Also, most of the data with $\varphi_A > +40\,\mathrm{deg}$ do not pass the quality controls, and therefore there are fewer data available to contribute to the antenna pattern. On the other hand, the H antenna exhibits an irregular pattern, with a sinusoidal - like behavior along all the $\varphi_A$ range. This behavior is consistent with the signal being affected by strong multipath.

The SNR pattern shows how the metallic structure affects the signal, but what we are really interested in is in the $\Delta\Phi$ pattern. This antenna pattern is shown in Fig. 6. We can also see how the $\Delta\Phi$ antenna pattern changed with respect to the original one measured in the anechoic chamber (e.g. Cardellach et al. (2014) Fig 5. - bottom panel). The fact that we set $\Delta\Phi = 0$ at 30 km makes the antenna pattern relative to that location. At 30 km height, the bending angle is small enough to consider that for a given azimuth, the elevation that corresponds to 30 km is very similar. Therefore, the antenna pattern characterizes the trends in the differential phase shift rather than the absolute values. The pattern of the phase difference arises from the combination of the patterns from the H and V antennas, and is irregular. The antenna pattern is used to correct every single $\Delta\Phi$ measurement, which is compared against the pattern for all the given ($\varphi_A, \theta_A$) as detailed in Sect. 5.

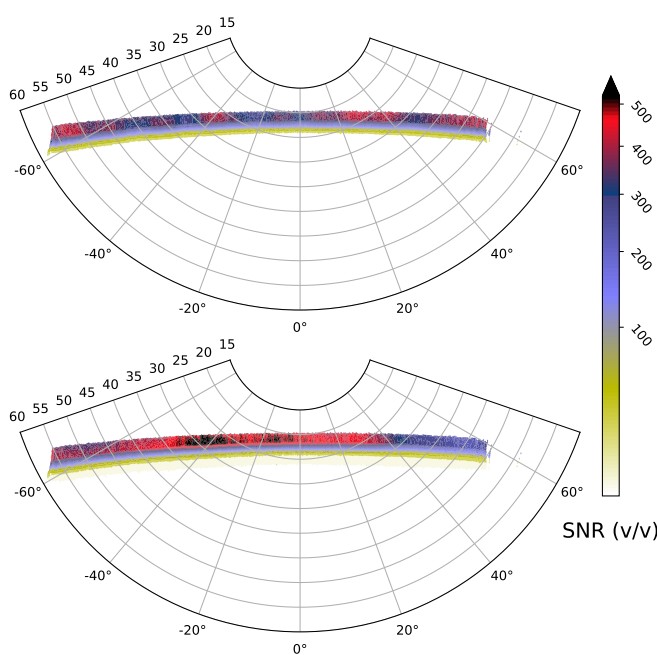

**Figure 5.** Effective antenna pattern for the Signal to Noise Ratio (colorscale) at the Horizontal port (top panel) and at the Vertical port (bottom panel), as a function of Azimuth (x axis) and Elevation (y axis) in the Antenna Reference frame ($\varphi_A, \theta_A$).

## 4 Assessment of the Ionospheric effect

Faraday Rotation($\Omega$) in the ionosphere can introduce a differential phase shift between the H and V components (Tomas et al., 2018). It depends on the electron density ($n_e$), the magnetic field ($\boldsymbol{B}$), and the relative orientation of the propagation direction ($\boldsymbol{r}$) and the magnetic field vector:

$$\Omega = \frac{-2.36 \cdot 10^4}{f^2} \int n_e(r)\, \boldsymbol{B} \cdot \boldsymbol{r}\, dr \tag{7}$$

where the constant is in International Units and the Faraday Rotation in radians. The Faraday Rotation induces a rotation of the polarization ellipse's axis described in linear basis. If the electromagnetic wave is perfectly Circularly Polarized, this rotation does not induce a differential phase shift. However, if the wave is not circularly polarized, the rotation induces a $\Delta\Phi$



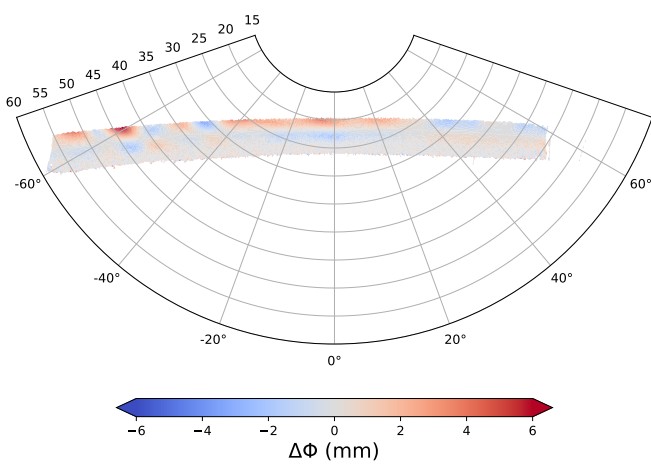

**Figure 6.** Antenna pattern for the $\Delta\Phi_m$ as a function of Azimuth (x axis) and Elevation (y axis) in the Antenna Reference frame ($\varphi_A, \theta_A$).

between the H and V components. There are two ways that can lead to a non-circularly polarized wave in a situation like the one we are analyzing here:

– Imperfect emission: Ideally, GPS satellites emit RHCP radio waves. However, it is not guaranteed that this emission is perfect and some impurities are to be expected. Therefore, if the emission is not perfect, radio waves travel through the ionosphere experiencing a $\Delta\Phi$ that is proportional to the Faraday Rotation (Tomas et al., 2018):

$$\Delta\Phi = -2m\sin(2\Omega + \Delta) \tag{8}$$

where $m$ and $\Delta$ characterize the difference of the emitted wave from the perfect circular polarization, i.e. $E = (1, me^{i\Delta})_{\{e_R, e_L\}}$.

– After crossing precipitation: When the radio wave crosses precipitation, even if it were perfectly Circularly polarized, it would experience a $\Delta\Phi$ induced by the hydrometeors. Therefore, after the rain in its way to the receiver, it crosses the ionosphere being non-circularly polarized. This implies that the second part of the ionosphere (i.e. the Faraday Rotation induced along the portion of the ionosphere that the ray crosses from its tangent point to the receiver, $\Omega_2$) induces a differential phase shift that depends on the $\Delta\Phi$ induced by precipitation (here identified as $\Delta\Phi_{\text{precip}}$):

$$\Delta\Phi = \left[1 - 2\Omega_2^2(t)\right]\Delta\Phi_{\text{precip}}(t) \tag{9}$$



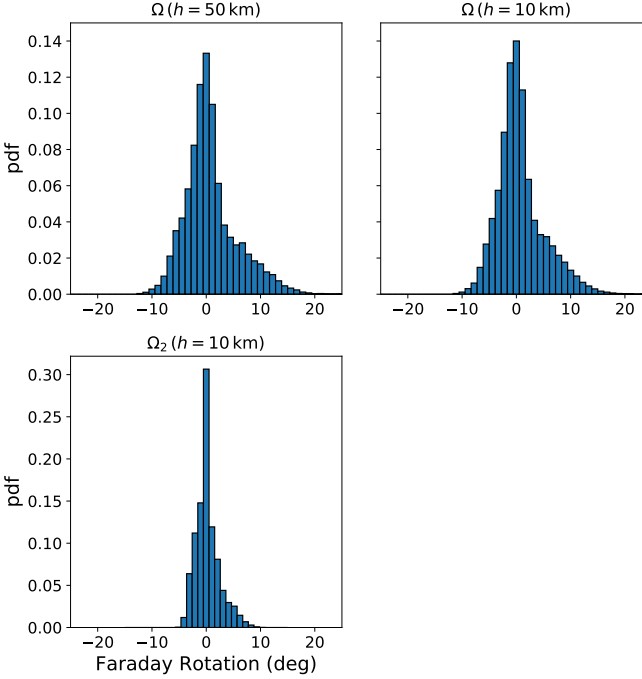

**Figure 7.** Histogram for the Faraday Rotation values. The top row represents the histogram for the total Farday Rotation ($\Omega$) evaluated at the ray with tangent point's height around 50km (left panel) and 10 km (right panel). In the bottom row there is the histogram for the second part of the Farday Rotation ($\Omega_2$), evaluated at the ray with tangent point's height 10 km.

Such expressions are thoroughly derived in Tomas et al. (2018). It is also shown in Tomas et al. (2018), based on simulations, that the effect of the Faraday Rotation due to the impurities in the emission (Eq. 8) should be possible to correct, and the effect after crossing rain (Eq. 9) is small enough to not introduce substantial errors in the measurements because $\Omega_2$ is generally low.

In the following section we analyze the observations based on the co-located electron density and magnetic field in order to
5 infer whether the ionosphere is inducing any noticeable $\Delta\Phi$.

### 4.1 Faraday Rotation for PAZ PRO events

First of all we need to know what are the typical values of Faraday Rotation along PRO rays. We take two heights, 10 and 50 km, at which we evaluate the Faraday rotation using the co-located values of $n_e$ and $B$ from IRI and IGRF. For every PRO, we compute the total Faraday Rotation at 50 and 10 km, and the second part of the Faraday Rotation at 10 km. The histogram for
10 all the cases is shown in Fig. 7

Summarizing Fig. 7, the observed values for the total Faraday Rotation has values between -12 and 20 deg, and between -6 and 10 for the portion of the Faraday Rotation between the tangent point and the receiver. We have also seen that the maximum difference of total Faraday Rotation between 50 and 10 km is as high as 3 deg, and as low as -2 deg. The difference of $\Omega$ between


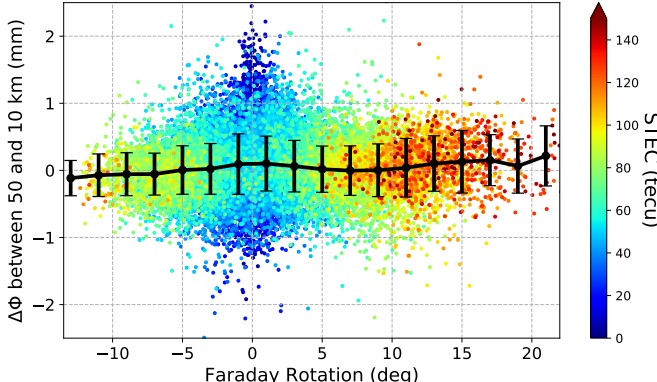

**Figure 8.** Trend of $\Delta\Phi(h)$ as a function of Faraday Rotation. The $\Delta\Phi$ is evaluated at the rays with tangent point's heights of 50 and 10 km, and the difference between them is plotted here as a function of $\Omega$ evaluated at the ray with tangent point's height of 50 km. The color of the points shows the integrated electron density content along the ray with tangent point's height of 50 km. Solid line is the mean and the error bars are the standard deviation.

two heights determines the trend in $\Delta\Phi$ that Faraday Rotation could be inducing, assuming that the wave is not perfectly circularly polarized when it crosses the ionosphere. In Fig. 8 we show the trend in $\Delta\Phi$, understood as $\Delta\Phi_{50\text{km}} - \Delta\Phi_{10\text{km}}$, as a function of the $\Omega$ at 50 km (which in its turn determines the trend in $\Omega$: the higher the $\Omega$, the higher the trend). We can see that the trend in $\Delta\Phi$ is imperceptible. This agrees with the simulations, which say that if there is a trend, it should be small (as high

as 0.6 mm) depending on $m$ and $\Delta$, that are unknown. In addition, $m$ and $\Delta$ should change by transmitter and probably by time and transmitter orientation, which makes them impossible to infer. The same study as in Fig. 8 has been done separating the data by GPS transmitter, with no revealing results.

Regarding the effect of the ionosphere after the rays have crossed precipitation (e.g. Eq. 9), we can evaluate the error introduced in our measured $\Delta\Phi$ with respect to $\Delta\Phi_{\text{precip}}$. With the values shown in Fig. 7, we obtain that the measured $\Delta\Phi$

is reduced by a 6% in the case of a $\Omega_2 \sim 10\deg$, which would be an extreme and rare situation. In the event that $\Omega_2 = 20\deg$, the measured $\Delta\Phi$ would be reduced a 25% below the actual $\Delta\Phi_{\text{precip}}$. The 90% of the observed $\Omega_2$ is confined between -3 and 4.7 deg, which implies that the measured $\Delta\Phi$ is reduced by a 1.34 %.

Based on the actual values for $\Omega$ and $\Omega_2$, it is safe to assume that the effect of the ionosphere into the $\Delta\Phi$ is generally negligible *(below the measurement noise level) and only in a very few cases can have a minor effect. This corroborates the

15 simulation study performed in Tomas et al. (2018) before the launch of the PAZ satellite. Nevertheless, the trend that is detected in $\Delta\Phi$ (measured between 50 and 10 km), small in average, is corrected from the whole observation, regardless whether it is a ionospheric effect or residuals from the first steps of the calibration of the observables.

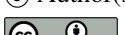



# 5    Calibration of the $\Delta\Phi$

We have gone through the different necessary steps before performing the calibration of the observables, from the acquisition of the signal to the antenna pattern characterization, and taking into account all the possible effects that can induce a differential phase shift besides precipitation. The steps followed to calibrate the $\Delta\Phi$ are identified and described in Fig. 9, and summarized
below:

- Acquisition of the signal: The incoming electromagnetic signal is collected at two independent linearly polarized antenna, orthogonal between them, oriented to get the horizontal and the vertical components of the radio wave, simultaneously. The difference between the excess phase of both ports (H and V) gives us the observable (step 1 in Fig. 9), which is further corrected for remaining cycle slips (step 3) and set to 0 in the regions above where any precipitation is possible (step
4). The observations are obtained as a function of time, but having precise information about the location and relative orientation of both the GPS and the PAZ satellites we can link time to azimuth and elevation from the receiving antenna point of view, so that we can obtain the measured differential phase shift as a function of such variables: $\Delta\Phi_m(\varphi_A, \theta_A)$. After the processing, time can also be linked to height, so we can have $\Delta\Phi_m(h)$ as well (step 2).

- Every PRO event is checked against precipitation (step 5). This allows us to group the events by rainy or non-rainy,
where rainy means that there exists precipitation inside the potentially sensed region whereas non-rainy means that no precipitation is present in the region. Furthermore, the co-located brightness temperature is used to further ensure that no precipitation was sensed by selecting those cases where the minimum $Tb$ is warmer than 250 K. Hence, PRO events are linked to $R$ and $Tb$.

- The ionospheric conditions (i.e. electron density) and the Earth's magnetic field (both intensity and orientation) are
evaluated at each ray's trajectory points for all the PRO observations, in order to compute the undergone Faraday Rotation (step 6). The total Faraday Rotation $\Omega$ , as well as the partial one (i. e. the Faraday Rotation suffered by the ray from the tangent point to the receiver) $\Omega_2$, are linked to all PRO.

At this point, every $i$th PRO event has some variables associated to it:

$$\Delta\Phi_m^i(t, \varphi, \theta, h), R^i, Tb^i, \Omega^i, \Omega_2^i$$

Data linked to no rain and low ionospheric activity are used to build the antenna pattern $\Delta\Phi_{\text{pattern}}(\varphi, \theta)$ (step 7). And this antenna pattern is used to correct the whole dataset of observations (step 8):

$$\Delta\Phi_c^i(\varphi, \theta) = \Delta\Phi_m^i(\varphi, \theta) - \Delta\Phi_{\text{pattern}}(\varphi, \theta), \tag{10}$$

where the subscript "c" stands for corrected.

The possible Faraday Rotation effect, although expected to be small in general (e.g. see 4), is not fully corrected by this
process. The antenna pattern characterization captures these ionosphere induced trends and possible errors induced by the





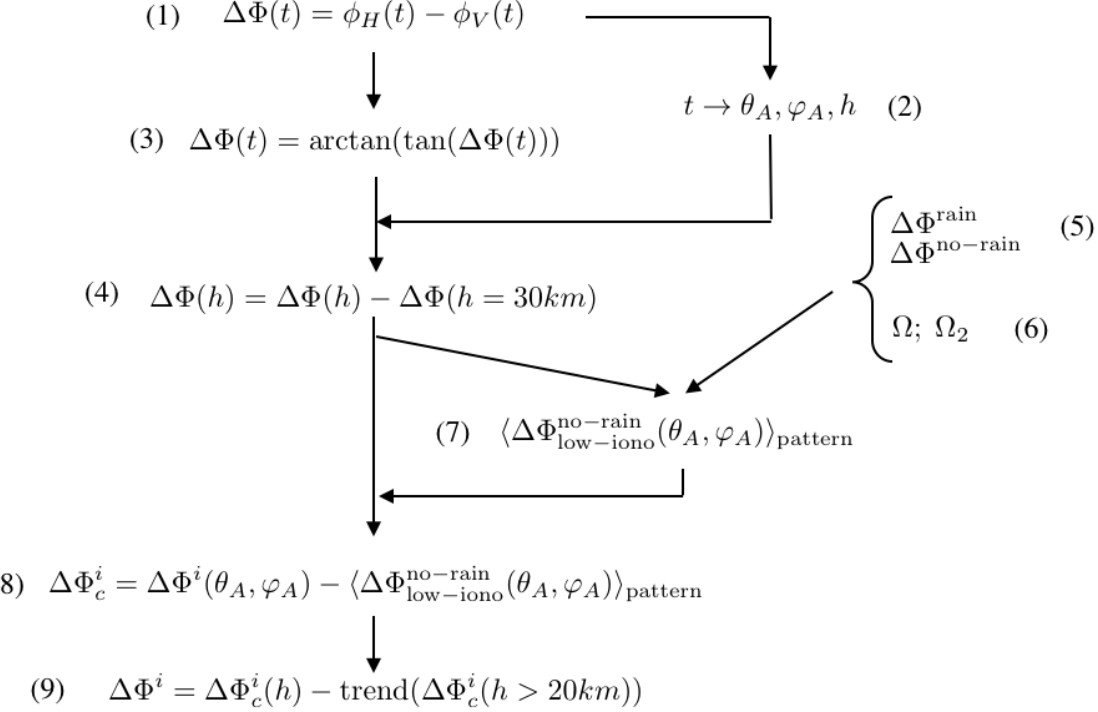

**Figure 9.** Block diagram identifying and describing the steps followed during calibration: (1) observable: phase difference between H and V; (2) correction for remaining cycle slips; (3) mapping time into other variables; (4) set the zero level at 30 km; (5) and (6) co-locations with precipitation information and ionospheric activity; (7) accumulation of free of rain and low ionospheric activity measurements to create the effective antenna pattern; (8) subtraction of the effective antenna pattern to each measurement; (9) correcting for remaining trends.

performance of the antenna. It is intentionally constructed with low ionospheric activity data so it does not capture the stronger trends induced by the active ionosphere, since they can be different and have nothing to do with the relative angle at which arrive to the antenna. Therefore, every $\Delta\Phi_c^i(\varphi, \theta)$ is corrected for remaining possible linear trends present above 20 km (step 9):

$$\Delta\Phi^i(h) = \Delta\Phi_c^i(h) - \text{trend}(\Delta\Phi_c^i(h > 20\,\text{km})) \tag{11}$$

where the linear trend is evaluated above 20 km and extrapolated to all heights. With this last step we obtain the calibrated $\Delta\Phi^i(h)$. After the whole calibration, we expect that $\Delta\Phi^i(h)$ is as similar to $\Delta\Phi_{\text{precip}}^i(h)$ as possible, where $\Delta\Phi_{\text{precip}}^i(h)$ is the differential phase shift induced only by precipitating hydrometeors. Note that the preliminary calibration in Cardellach et al. (2019) did not include steps 5 to 8.





## 5.1 Smoothing of the signal

Once we have calibrated the signal, we smooth it to reduce the uncertainty. PRO are acquired at 50 Hz, but for the purposes of detecting precipitation it is enough to have measurements at 1 s resolution. While the smoothing reduces the standard error by accounting for more samples for each measurement (e.g. we use 50 points obtained at 50 Hz to represent the measurement at 1 s resolution), its counterpart is that it reduces the vertical resolution of the observation, being of about few hundred meters after smoothing. Generally, a simple running average window would be applied to perform the smoothing. However, here we want to stress the fact that the measurements with higher Signal to Noise Ratio have less uncertainty. The uncertainty in the phase measurement is determined by the SNR of each measurement (e.g. Cardellach et al., 2014), so the uncertainty of the $\Delta\Phi$ comes from the propagation of such error from both H and V ports. Therefore, instead of a simple average, here we perform a 1 second weighted average where the weight is represented by the SNR value, so that values of $\Delta\Phi$ associated to higher SNR contribute more than those associated to lower SNR. In this case, since we are combining both the measurements from the H and V ports, the SNR that we use for the weighted average is: $\mathrm{SNR} = (\mathrm{SNR}_H + \mathrm{SNR}_V)/\sqrt{2}$. The SNR values are limited so that only those above 10 V/V contribute to the mean.

## 6 Validation of the $\Delta\Phi$

The smoothed calibrated measurements $\langle\Delta\Phi\rangle_{1\mathrm{s}}$ are to be validated against IMERG. For comparison and standardization purposes, we interpolate the $\langle\Delta\Phi\rangle_{1\mathrm{s}}$ into a 100 m grid spacing profiles from 0 to 30 km: $\Delta\Phi(\mathrm{h}_{100\mathrm{m}})$. For these profiles we can perform the mean and the standard deviation at each altitude. First of all, we group them by their linked precipitation and brightness temperature: (1) no-precipitation ($R = 0\,mm/h$ and $Tb > 250\,K$), precipitation ($R > 0.1\,mm/h$), and heavy precipitation ($R > 1\,mm/h$). For these three groups we compute the mean and standard deviation as a function of height. In Fig. 10 we show the results. We can see how for the no-precipitation group, the $\Delta\Phi(\mathrm{h}_{100\mathrm{m}})$ averages to 0 for the whole vertical profile (by design), and the standard deviation ($\sigma_{\Delta\Phi}(h)$) increases with decreasing altitudes. In the right panel of Fig. 10 we show a more detailed profile of the $\sigma_{\Delta\Phi}(h)$.

The first remark is that $\sigma_{\Delta\Phi}(h) = 1.5\,\mathrm{mm}$ at 2.5 km of height. This is better than the study performed in Cardellach et al. (2019), expected since here we calibrated the signal using the antenna pattern and we have performed the weighted average smoothing. It is also very close to the theoretically predicted sensitivity in Cardellach et al. (2014). The second noticeable feature is the peak in $\sigma_{\Delta\Phi}(h)$ around 7 km. This feature is due to instrumental effects in some occultations near the closed loop to open loop transition, and is an open issue under investigation. Still in Fig. 10, the precipitation and heavy precipitation groups (blue and red, respectively) exhibit large positive values below 10 km, although positive values start to be noticeable below 15 km. The positive peaks are well above the standard deviation of the no precipitation group, indicating sensitivity to precipitation and consistent with Cardellach et al. (2019).

As it was done in Cardellach et al. (2019), we can characterize each PRO observation to a single value derived from the $\Delta\Phi(\mathrm{h}_{100\mathrm{m}})$. This is done by averaging $\Delta\Phi(\mathrm{h}_{100\mathrm{m}})$ between two different heights. In this case, we use 0 to 10 km, obtaining $\langle\Delta\Phi\rangle_{0-10\mathrm{km}}$. To associate one single measurement is useful to validate the observations against the precipitation products. In





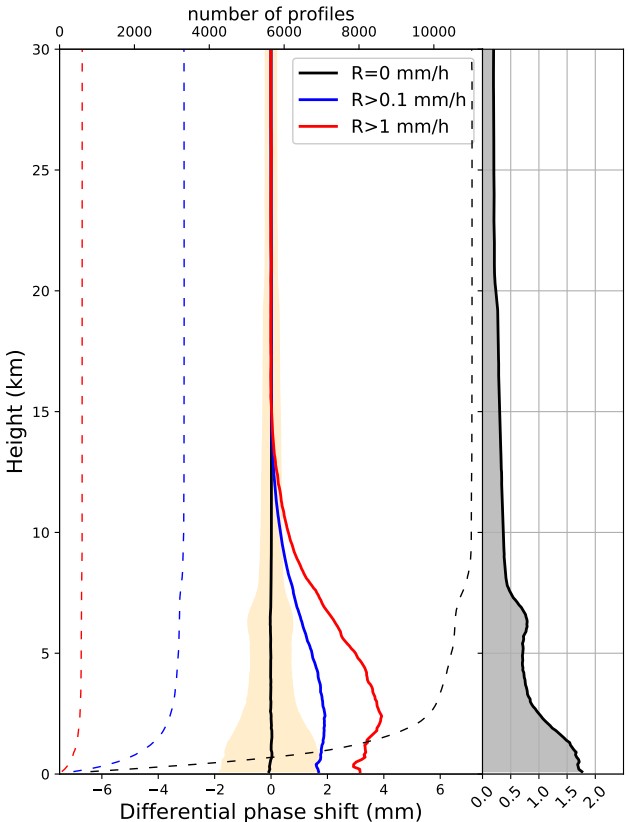

**Figure 10.** Mean (solid black line) and standard deviation (orange shaded) of $\Delta\Phi(\mathrm{h}_{100m})$ as a function of height for the PRO events collected under no rain conditions. The solid blue and red lines show the mean of $\Delta\Phi(\mathrm{h}_{100m})$ as a function of height for the PRO events collected under $R > 0.1\,\mathrm{mm/h}$ and $R > 1\,\mathrm{mm/h}$, respectively. The dashed lines show the number of collected profiles (top axis) as a function of height corresponding to each group (no precipitation, $R > 0.1\,\mathrm{mm/h}$ and $R > 1\,\mathrm{mm/h}$). The right side panel shows a more detailed vertical profile of the standard deviation $\sigma_{\Delta\Phi}(h)$ for the PRO events collected under no rain conditions (gray shaded area).

Fig. 11 we show the $\langle\Delta\Phi\rangle_{0-10km}$ as a function of the associated rain rate. The binned mean (solid line) show how $\langle\Delta\Phi\rangle_{0-10km}$ tend to increase as $R$ increases, exhibiting sensitivity to the intensity of precipitation.

It is also interesting to assess the percentage of cases that exceed a certain threshold of $\langle\Delta\Phi\rangle_{0-10km}$ given a precipitation value. This sets a detectability metrics of $\langle\Delta\Phi\rangle_{0-10km}$ based on the co-locations, and we can assess the quantity of false positives. We show the results in the top panel of Fig. 12. In this plot we see how for no precipitation, there is a 10% of cases that exceed $\langle\Delta\Phi\rangle_{0-10km} = 0.5\,\mathrm{mm}$, while there are almost no cases exceeding 1 mm (or higher). This tells us that the rate of false positives is very low, and depending on the threshold we choose, is almost non existent. The same plot shows the percentage of cases exceeding different thresholds of $\langle\Delta\Phi\rangle_{0-10km}$ (represented by different colors, see inset legend). For





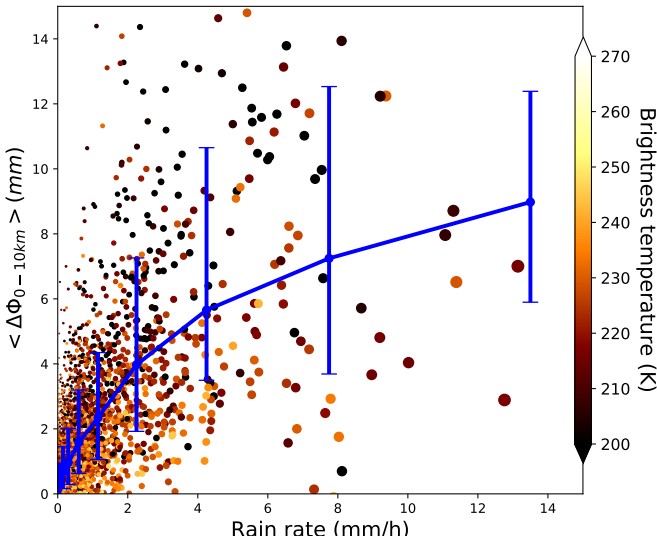

**Figure 11.** $\langle\Delta\Phi\rangle_{0-10\mathrm{km}}$ as a function of rain rate. The color indicates the minimum $Tb$ for every case. The solid blue line represents the mean of all the data inside precipitation bins. The errorbars comprise the 85% of the data.

example, as we can see in Table 1, 81% of the cases exceed $\langle\Delta\Phi\rangle_{0-10\mathrm{km}} = 1\,\mathrm{mm}$ when precipitation is heavier than 1 mm/h, and how more than 80% of the cases exceed $\langle\Delta\Phi\rangle_{0-10\mathrm{km}} = 2\,\mathrm{mm}$ when precipitation exceeds 5 mm/h.

In the same way, we can assess which percentage of cases exceeds certain precipitation given a $\langle\Delta\Phi\rangle_{0-10\mathrm{km}}$. This is shown in the bottom panel of Fig. 12. In this way we can assess the false negatives and see how likely is to detect precipitation given

a $\langle\Delta\Phi\rangle_{0-10\mathrm{km}}$. The leftmost region of the plot shows the fraction of cases exceeding certain precipitation when the observed $\langle\Delta\Phi\rangle_{0-10\mathrm{km}}$ is small (i.e. smaller than 0.1 mm). For a precipitation threshold of 0.01 mm/h, this fraction is around a 15%, while for precipitation heavier than 1 mm/h (and higher) is almost 0 (see Table 1). We can also see how for example, when the measured $\langle\Delta\Phi\rangle_{0-10\mathrm{km}}$ is larger than 1 mm, there is a 73.6% chance of measuring precipitation with 0.1 mm/h or higher.

## 6.1  Variability by Transmitter

As it has been mentioned in Sec. 4.1, the way the signal is emitted from the GPS transmitter can also have an effect into $\Delta\Phi$. In particular, if the emission is not perfectly RHCP. However, this effect should be small (e.g. see Fig. 8). Here we investigate whether different transmitters have similar statistics (as we expect), or not. To do so we reproduce the analysis done to generate Fig. 10 grouping the data by transmitter. The results for the $\Delta\Phi(\mathrm{h}_{100\mathrm{m}})$ and $\sigma_{\Delta\Phi}(h)$, evaluated at 3 km of altitude, are shown in Fig. 13.

The results for the different transmitters (also separated here by Block, i.e. the version of satellite) show how the $\Delta\Phi(\mathrm{h}_{100\mathrm{m}})$ and $\sigma_{\Delta\Phi}(h)$ are consistent with the global mean and $\sigma$, showing no dependence on the transmitter.





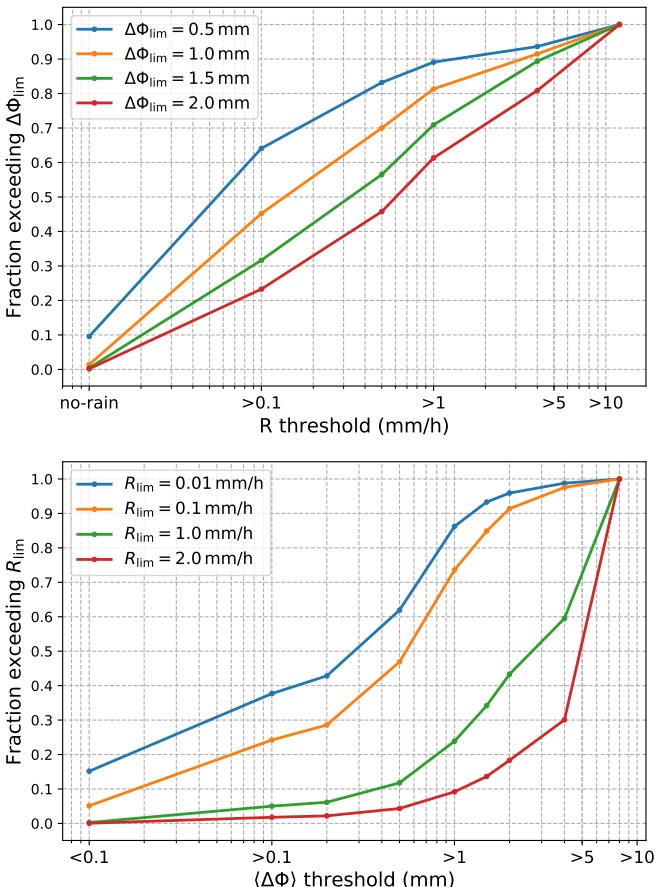

**Figure 12.** (Top) Fraction of cases exceeding certain $\langle \Delta\Phi \rangle_{0-10\mathrm{km}}$ value (blue: 0.5 mm; orange: 1.0 mm; green: 1.5 mm; and red: 2.0 mm) as a function of the associated precipitation threshold. (Bottom) Fraction of cases exceeding a certain precipitation value (blue: 0.01mm/h; orange: 0.1 mm/h; green: 1.0 mm/h; and red: 2.0 mm/h) as a function of measured $\langle \Delta\Phi \rangle_{0-10\mathrm{km}}$ threshold.

## 6.2 Cross polarization isolation

The metallic structure could have worsened the overall performance of the polarimetric antenna by reducing the cross polarization isolation. Some simulations on how it affects the $\Delta\Phi_{\mathrm{precip}}$ with respect to the measured one are performed in order to assess this effect. The transmission matrix that represents the antenna can be expressed as:

$$
\begin{bmatrix} E_h \\ E_v \end{bmatrix} = \begin{bmatrix} a_{hh} & a_{hv} \\ a_{vh} & a_{vv} \end{bmatrix} \begin{bmatrix} E_h^i \\ E_v^i \end{bmatrix}
\tag{12}
$$



**Table 1.** Summary of Fig. 12 for some representative thresholds.

| Rain threshold | % cases exceeding $\Delta\Phi=$ | | | | $\Delta\Phi$ threshold | % cases exceeding $R=$ | | | |
|---|---|---|---|---|---|---|---|---|---|
| | 0.5 mm | 1.0 mm | 1.5 mm | 2.0 mm | | 0.01 mm/h | 0.1 mm/h | 1 mm/h | 2 mm/h |
| no rain | 9.6 % | 1.4 % | 0.4 % | 0.1 % | $\Delta\Phi < 0.1$ mm | 15.2% | 5.1% | 0.2% | 0.0% |
| $R > 0.1$ mm/h | 64.1% | 45.2% | 31.7% | 23.3% | $\Delta\Phi > 0.1$ mm | 37.7% | 24.2% | 5.0% | 1.8% |
| $R > 1$ mm/h | 89.1% | 81.3% | 70.9% | 61.3% | $\Delta\Phi > 1$ mm | 86.2% | 73.6% | 23.9% | 9.1% |
| $R > 5$ mm/h | 94.4% | 92.6% | 90.7% | 83.2% | $\Delta\Phi > 2$ mm | 95.9% | 91.4% | 43.3% | 18.3% |

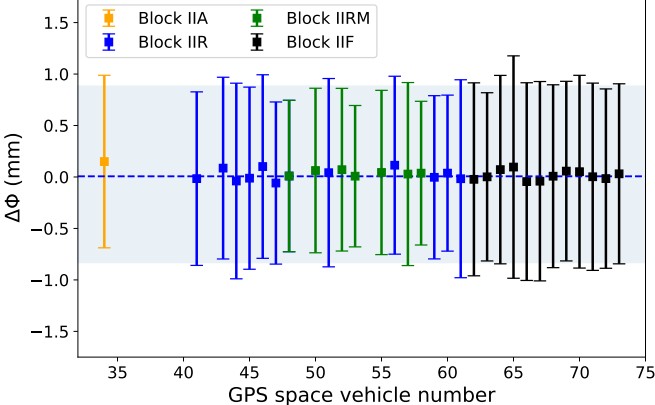

**Figure 13.** Mean and standard deviation of the $\Delta\Phi$ for the rain free cases, at a height of 3 km, as a function of the transmitter (i.e. the GPS space vehicle number). The blue dashed line and the light blue shadow represent the mean and standard deviation of the whole dataset of free rain cases. Different colors for the error-bar points represent different gps satellite Blocks, as indicated in the legend.

For a good cross polarization isolation (e.g. $< -30$ dB), the terms $a_{hv}$ and $a_{vh}$ can be approximated to 0. In this case, the cross polarization isolation is not known due to the disturbance included by the metallic structure, but simulations provided by Hisdesat suggest a cross polarization isolation between $-15$ and $-20$ dB, which means that $a_{hv}$ and $a_{vh}$ cannot be neglected.

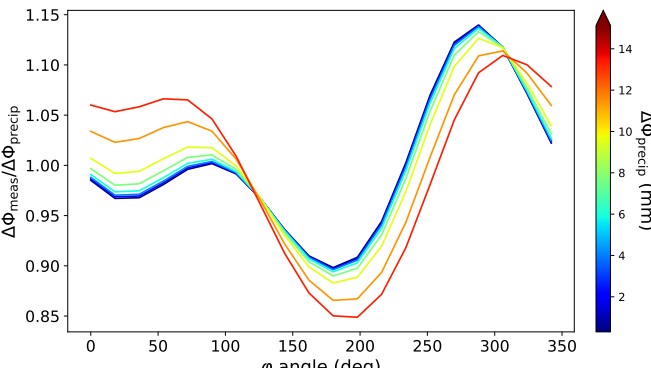

**Figure 14.** Ratio between the measured $\Delta\Phi$ and the precipitation induced $\Delta\Phi_{\mathrm{precip}}$ as a function of the complex angle $\varphi$ in $ae^{i\varphi}$, for a=17 and for different values of precipitation. The value of $\Omega_2 = 10\,\mathrm{deg}$. The precipitation values range between an induced $\Delta\Phi_{\mathrm{precip}}$ of 0.1 mm (bluer) to 15 mm (redder).

We have performed simulations assuming $a_{hv} = a_{vh} = ae^{i\varphi}$, where $a = 0.17$ (corresponding to $-15$dB, i.e. conservative approach) and that $\varphi$ can take any value. We compare the $\Delta\Phi$ that we would measure without the antenna with the $\Delta\Phi$ if the antenna is present and introduces a differential phase shift between the H and V components from the poor cross polarization isolation. The simulations are performed accounting different $\Delta\Phi_{\mathrm{precip}}$ and different values for $\Omega_2$. The results, plotted in

Fig. 14, show that the ratio between the measured $\Delta\Phi$ and the actual $\Delta\Phi_{\mathrm{precip}}$ can be as high as 15% (only the results for $\Omega_2 = 10$ deg are shown). However, the maximum variance comes from the variation of $\varphi$, which is unknown and probably not constant. Averaging over all the results for different $\varphi$, the average ratio is 1, and the standard deviation is around a 7%. It is also important to notice that the variability induced by $\Omega_2$ are also included, therefore the values of the ratio include both the ionosphere effect and the poor cross polarization isolation.

**7   Conclusions**

In this manuscript we have described the steps and the procedure followed to calibrate and validate the PRO $\Delta\Phi$ observable. The calibration of the observable is a critical step of the mission that has to ensure the quality and robustness of the observables. Being the first time that these kind of measurement are being obtained, the validation of the observables is also very important, since it will establish a reference for future missions.

First of all, the calibration of the signal has been performed by using the existing data to characterize the antenna pattern. Such an exercise is more important than it should be due to the interferences introduced by a metallic adapter that had to be installed above the antenna, to adapt the satellite to a new launcher. In order to not introduce features coming from the kind of signals that we aim to detect (i.e. precipitation induced $\Delta\Phi$), the characterization is performed using only data collected in free of rain scenarios. Furthermore, ionosphere could introduce small $\Delta\Phi$ through the Faraday Rotation, hence observations that





have sensed regions with high ionospheric activity are also discarded for the calibration. Finally, the antenna pattern is then used to correct all the observations, regardless of precipitation or ionospheric activity.

The corrected observations are thoroughly validated. First, we have performed the statistical analysis of both no-precipitation and precipitation groups of observations. The mean and standard deviation of the no-precipitation profiles set the quality of

the observations. Without the presence of precipitation, what remains are the uncertainties and the un-sought effects, so the standard deviation tell us the noise level of the measurement. Inside the noise level we assume that we can have the thermal noise arising of the phase measurements, residual effect from the calibration, and cross polarization terms from the non-perfect isolation of the antenna. In spite of that, the vertical profile of the standard deviation (see Fig. 10) shows a good noise level (below 1.5 mm above 2 km, below 1 mm above 3 km, and better than 0.5 mm above 8 km), close to what was predicted in the

initial sensitivity studies for the experiment (Cardellach et al., 2014). It is also confirmed that the Faraday Rotation effect into the final observable is small, and that the transmitter polarization impurities are negligible.

In addition, the mean $\Delta\Phi$ as a function of height for the precipitation groups exhibit a clear and distinguishable positive peak, reaching altitudes above 10 km and exceeding the $\Delta\Phi = 5\,\mathrm{mm}$ in the lower layers for the group comprising the precipitation rates larger than 1 mm/h. This clearly indicates that the measurement is sensitive to precipitation, corroborating the initial

findings in Cardellach et al. (2019). It also indicates that the measurement might be sensitive to higher altitude phenomena other than precipitation, such as ice or melting particles, usually present above the freezing level and particularly in the heavy tropical precipitation structures.

Further validation is performed using the vertical average of $\Delta\Phi(\mathrm{h}_{100\mathrm{m}})$ between the surface level and 10 km for each individual observation, defined as $\langle\Delta\Phi\rangle_{0-10\mathrm{km}}$. This allows us to use single values rather than vertical profiles associated to

each observation, for simplicity in the validation process. Using this approach we can assess the variation of $\langle\Delta\Phi\rangle_{0-10\mathrm{km}}$ with increasing $R$ (see Fig. 11). The fact that $\langle\Delta\Phi\rangle_{0-10\mathrm{km}}$ keeps increasing as $R$ increases tells us that $\Delta\Phi$ measurements are not only sensitive to precipitation, but also to its intensity. Here we want to remind that in this context precipitation intensity means higher mean rain rate integrated for the sensed region (see Sect. 2.3), which could either mean more intense precipitation or a larger precipitation cell, or both.

The same $\Delta\Phi(\mathrm{h}_{100\mathrm{m}})$ measurement is used to evaluate the detectability of precipitation for different thresholds (see Fig. 12). For example, we can state that more than a 80% of the cases with $R > 1\mathrm{mm/h}$ exceed $\Delta\Phi(\mathrm{h}_{100\mathrm{m}}) = 1.5\mathrm{mm}$. In a different, yet equivalent, way we can state that the 50% of the cases with $\Delta\Phi(\mathrm{h}_{100\mathrm{m}}) > 2\mathrm{mm}$ exceed a $R = 1\mathrm{mm/h}$, but more than the 90% will have $R > 0.1\mathrm{mm/h}$. Therefore, the *detectability* will depend on the threshold that one sets. On the other hand, the same study shows low values for false positives and false negatives regardless of the chosen threshold. Setting the thresholds

towards the heavier rain range (although heavy rain is not qualitatively defined here) decreases the false positives and negatives dramatically, exhibiting a very good performance of the technique in detecting rain. It is important here to emphasize the fact that we are evaluating the performance in detecting rain rather than quantifying its rate, and the validation in the context of this paper confirms that capability.





*Competing interests.* The authors declare that they have no conflict of interest.

*Acknowledgements.* R. Padullés research was supported by an appointment to the NASA Postdoctoral Program at the Jet Propulsion Laboratory, administered by Universities Space Research Association under contract with NASA. The JPL co-authors acknowledge support from the NASA US Participating Investigator (USPI) program. The work conducted at ICE-CSIC/IEEC was supported by the Spanish grant
5   ESP2015-70014-C2-2-R. Part of Cardellach's contribution has been supported by the Radio Occultation Meteorology Satellite Application Facility (ROM SAF) which is a decentralized operational RO processing centre under EUMETSAT.



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
