# Peer review of "Calibration and Validation of the Polarimetric Radio Occultation and Heavy Precipitation experiment Aboard the PAZ Satellite"

_Atmospheric Measurement Techniques, 2019_

## Referee Comment (RC1) · Anonymous Referee #1 · 14 Oct 2019

**Review of paper "Calibration and Validation of the Polarimetric Radio Occultation and Heavy Precipitation experiment Aboard the PAZ Satellite" by Ramon Padullés, Chi O. Ao, F. Joseph Turk, Manuel de la Torre Juárez, Byron Iijima, Kuo Nung Wang, and Estel Cardellach**

**General Remarks**

The paper presents impressive results on the calibration and validation of polarimetric radio occultation observation acquired by PAZ satellite. The authors thoroughly analyze the factors that can have an influence upon the observable phase differences characterizing the precipitation. Still, there are some concerns, especially regarding the phase ambiguity removal. Although I believe that this should significantly worsen the results, still the authors should address these concerns.

**Specific Comments**

Page 2, lines 5–6: *The fact that in the PAZ satellite the incoming electromagnetic field is acquired at two linear and orthogonal polarizations allow us…*
The fact … allows …

Page 4, lines 19–21: *Even though the initial processing of the raw data corrects for cycle slips (i.e. changes in $\phi$ of more than one cycle in 20 consecutive measurements), after computing $\Delta\Phi(t)$ some jumps in the observable are detected. These jumps are associated to cycle slips that remained uncorrected before, or appeared after computing the difference between the two $\phi(t)$ (h and v).*
This is not clear. Why should any cycle slips remain uncorrected? Should not there be any physical cause for this effect? It should be better to present some examples.

Page 4, lines 22–26: *Therefore, the $\Delta\Phi(t)$ is also corrected for cycle slips in the following way:*

$$\Delta\Phi(t) = \arctan\left(\tan\left(\Delta\Phi(t)\right)\right) \tag{2}$$

*This approach should correct the cycle slips remaining in the data. However, this approach can still introduce a $2\pi$ jump in the phase if this is too close to $\pm\pi/2$, although this is an infrequent situation. Since the $\Delta\Phi(t)$ tends to follow a rather smooth variation in the presence of precipitation, events inducing such large $\Delta\Phi(t)$ values can be easily identified and treated accordingly.*
This paragraph raises some concerns. First, because arctan function varies from $-\pi/2$ to $\pi/2$, maximum what it can introduce is a jump by $\pi$. Second, why using formula (2), which does not distinguish between $\Delta\Phi$ values that differ by $\pi$? Referring to the fact that this is an "infrequent situation" does not really help, because the expense of implementing the standard procedure of the evaluation of the accumulated phase is low.

Page 4, lines 27–29: *For each port, data are processed to obtain $N(h)$. To assign a height to each time measurement (e.g. excess phase or SNR) is complicated, specially when atmospheric multipath is present. To do so we rely on the inverse Abel transform and we assign a tangent height (the height of the tangent point of each ray) to each phase and SNR measurement $\Delta\Phi(h_t)$, and $SNR(h_t)$.*
Provide more detail on your procedure of the evaluation of $h_t$. What about multipath propagation? How should you treat the multi-valued dependence $h_t(t)$ in this case?
Replace "specially" with "especially".

Page 7, Figure 3, caption: *… more portion of the rays happen to be below the set altitude thresholds …*

Re-write this, e.g. as follows: … longer ray segments reach below the altitude thresholds…

Page 8, lines 10–15: *To account for the relative orientation, we use the quaternions, provided along with the orbits data. … To account for the satellite orientation and maneuvers we use the quaternions, that precisely define the orientation of the satellite with respect to the center of the Earth Centered reference frame at any given time.*
Quaternions are briefly mentioned here and never more arise in the paper. Quaternions provide one of different representations of the spatial orientation, and they are hardly a unique means for solving the problem in question. They are, therefore, either unnecessary to mention or deserve a deeper explanation.

Page 9, line 4: *The positive values of $\phi_A$ are defined such as the angle increases towards the positive $Y$.*
Clarify the sentence, e.g. like this: *Angle $\phi_A$ has the same sign as $Y$.*

Page 17, line 9–11: *Therefore, instead of a simple average, here we perform a 1 second weighted average where the weight is represented by the SNR value, so that values of $\Delta\Phi$ associated to higher SNR contribute more than those associated to lower SNR.*
Because SNR and $\Delta\Phi$ are correlated random processes, should not this result in a systematic error?

---

## Referee Comment (RC2) · Anonymous Referee #2 · 22 Oct 2019

The manuscript presents the calibration and validation studies for the Radio Occultations and Heavy Precipitation experiment aboard the PAZ satellite. I find the study interesting and well presented. I only have a few minor points:

Abstract: please write in words what the 'dphi observable' is.

Page 4, L1: "The JPL designed IGOR+ receiver installed in PAZ collects RO data at a rate of 50Hz. Each RO is tracked independently in the two ports dedicated to the H and V polarized antennas. Therefore, each port output is processed independently. Raw phase data ..." please write Integrated GPS Occultation Receiver (IGOR) and provide some more details. Tracking modes, setting and rising occultations, etc.?

[Figure]

Page 4, L27:"For each port, data are processed to obtain N(h). To assign a height to each time measurement (e.g. excess phase or SNR) is complicated, specially when atmospheric multipath is present. To do so we rely on the inverse Abel transform and we assign a tangent height (the height of the tangent point of each ray) to each phase and SNR measurement, (ht) and SNR(ht) . As a convention, the height that is assigned to each time is the mean of the heights obtained in the H and V ports at that time..." Are the heights obtained from the H and V ports very different? Are the BA (N) profiles very different? Can you provide an example (maybe in Fig.2 to the right).

Page 5, L3: "The whole processing is applied to 59,704 occultations, of which a total of 42,209 pass through the JPL quality control. The quality control is passed if the retrieved refractivity profiles between 0 and 30 km (for both H and V) are within 10% of the co-located NCEP Global Forecast System (GFS). Those that do not pass the quality control are discarded." I understand that the focus of this paper is not N profiles, but maybe you could add some N statistics here or later. Also see comment regarding Fig.10 below.

Page 5, L10:"The ray tracing uses the actual retrieved refractivity profile to account for the bending of the rays." I suggest to remove the word 'actual'.

Page 13, L8:"We take two heights, 10 and 50 km, at which we evaluate the Faraday rotation using the co-located values of ne and B from IRI and IGRF" I suggest to add in brackets the corresponding year. whenever the ionosphere comes into play at least the year (2018/2019) should be mentioned. Maybe you could also mention here or later if you expect some impact during high solar activity.

Page 14, L12:"...which implies that the measured is reduced by a 1.34 %." remove 'a' here and i suggest to write 1.3% instead of 1.34%.

Page 18, Fig10.: I am curious to see some N statistics. In a recent study

Padullés, R., Cardellach, E., Wang, K.-N., Ao, C. O., Turk, F. J., and de la Torre-

Juárez, M.: Assessment of global navigation satellite system (GNSS) radio occultation refractivity under heavy precipitation, Atmos. Chem. Phys., 18, 11697–11708, https://doi.org/10.5194/acp-18-11697-2018, 2018.

you presented interesting results and concluded:

'...This is the aim of polarimetric radio occultations, which will provide joint products of temperature, pressure, and moisture and an indication of the amount of precipitation (mostly sensitive to the heaviest) at each vertical level (Cardellach et al., 2017) with the objective of advancing the understanding of heavy precipitation events, closely linked with high specific-humidities conditions...'

Would it be too much of an effort to add to the right the corresponding N statistics (w.r.t. say the NCEP GFS) ? I would like to see if the positive mean deviation correlates with the positive N bias.

Page 19, L1:" as we can see in Table 1, 81% of the cases exceed" write 1.8% instead of 1.81%.

Conclusion, L31: " It is important here to emphasize the fact that we are evaluating the performance in detecting rain rather than quantifying its rate, and the validation in the context of this paper confirms that capability" but this can be also done with the IMERG product? Can you write something about the potential benefit of your product here or in the introduction.

---

## Author Comment (AC1) · 3 Feb 2020

**Response to Reviewer 1**

First of all we want to thank the time spent by the reviewers in providing comments and corrections to this manuscript, that are going to greatly improve its quality. Comments from the reviewers made us notice some mistakes in the processing of the data, that have been corrected. Therefore, the data processing has been repeated and more data have been included. All figures have been replaced with the new versions, and even though some might seem very similar, they all now include the whole dataset. The main results have not changed. Few numbers have been changed, with no implications for the conclusions.

Below we specifically address the reviewer's comments one by one. Please note that the text in blue is the original reviewer's comment and in black is our answer.

**General Remarks**

The paper presents impressive results on the calibration and validation of polarimetric radio occultation observation acquired by PAZ satellite. The authors thoroughly analyze the factors that can have an influence upon the observable phase differences characterizing the precipitation. Still, there are some concerns, especially regarding the phase ambiguity removal. Although I believe that this should significantly worsen the results, still the authors should address these concerns.

**Specific Comments**

1. Page 2, lines 5–6: *The fact that in the PAZ satellite the incoming electromagnetic field is acquired at two linear and orthogonal polarizations allow us…* The fact … allows …

   Corrected.

2. Page 4, lines 19–21: *Even though the initial processing of the raw data corrects for cycle slips (i.e. changes in $\phi$ of more than one cycle in 20 consecutive measurements), after computing $\Delta\phi(t)$ some jumps in the observable are detected. These jumps are associated to cycle slips that remained uncorrected before, or appeared after computing the difference between the two $\phi(t)$ (h and v).*
   This is not clear. Why should any cycle slips remain uncorrected? Should not there be any physical cause for this effect? It should be better to present some examples.

   Cycle slips may remain uncorrected when the signal is too noisy. H and V are tracked independently through different RF channels and could have uncorrected cycle slips occurring at different times. When differentiating the phase in the H port and the phase in V port, such cycle slips may become more evident.

3. Page 4, lines 22–26: *Therefore, the $\Delta\phi$ is also corrected for cycle slips in the following way:*

   $$\Delta\phi = arctan(tan(\Delta\phi(t))) \qquad (2)$$

   *This approach should correct the cycle slips remaining in the data. However, this approach can still  introduce a 2π jump in the phase if this is too close to +-π/2, although this is an infrequent situation. Since the $\Delta\phi$ tends to follow a rather smooth variation in the presence of*

*precipitation, events inducing such large Δϕ values can be easily identified and treated accordingly.*
*This paragraph raises some concerns. First, because arctan function varies from -π/2 to +π/2, maximum what it can introduce is a jump by π.*

We agree, this was a typo.

Second, why using formula (2), which does not distinguish between Δϕ values that differ by π? Referring to the fact that this is an "infrequent situation" does not really help, because the expense of implementing the standard procedure of the evaluation of the accumulated phase is low

What the reviewer points out is true during the open-loop tracking. We acknowledge the reviewer for having noticed this error. The processing chain has been changed to account for that, so that the open loop region is corrected using:

$$Δϕ = arctan2(sin(Δϕ), cos(Δϕ))$$

The reviewer will note that most of the figures, tables and values shown in the manuscript have slightly changed after reprocessing the entire data set. The new results are nevertheless very similar to the previous ones, so, fortunately, the error did not have a large impact on the results. The manuscript now includes the following sentence:

During closed-loop (CL) tracking that occurred above ~ 8-10 km altitude, the phase is obtained with half-cycle ambiguity [Ao et al. 2003]. During open-loop (OL) tracking that occurred below ~ 8-10 km altitude, the tracking data are processed on the ground with the 50 Hz navigation modulation removed that enables full-cycle phase reconstruction [Ao et al. 2009]. Therefore, we correct for half-cycle slips during CL and full-cycle slips during OL.

Furthermore, Figure 2 shows an example of the two kind of cycle slip corrections, and Figure 9 now includes distinctive OL and CL cycle slip corrections in the diagram.

4. Page 4, lines 27–29: *For each port, data are processed to obtain N(h). To assign a height to each time measurement (e.g. excess phase or SNR) is complicated, specially when atmospheric multipath is present. To do so we rely on the inverse Abel transform and we assign a tangent height (the height of the tangent point of each ray) to each phase and SNR measurement Δϕ($h_t$), and SNR($h_t$).*
Provide more detail on your procedure of the evaluation of $h_t$ . What about multipath propagation? How should you treat the multi-valued dependence $h_t$(t) in this case?

As the reviewer correctly pointed out, the association of height with time became ambiguous when atmospheric multipath was present. This was estimated in our processing as follows. Using geometric optics (GO) retrievals, we obtain an approximated relationship between the impact parameter and received time (a_{GO}(t)). This is then used to map the canonical-transformed impact parameter a_{CT} to a unique time t. We are aware that this is not exact since the relation a_{GO}(t) contains errors under atmospheric multipath conditions. We have estimated the uncertainty in height determination using the GO retrieved bending angles, which gave fluctuating, non-montonously varying impact parameter with respect to time in the presence of atmospheric multipath. Based on the variation of impact parameter, we computed

the RMS height variation for one week of PAZ occultation data. The figure below shows that the height uncertainty due to atmospheric multipath varies from 0.1 km at 6 km altitude to 0.6 km at 2 km altitude in the tropics.

We explain this in the paper, but we do not include the figure. It is included here (below) as reference for the reviewer.

[Figure]

5. Replace "specially" with "especially".

   Changed.

6. Page 7, Figure 3, caption: … *more portion of the rays happen to be below the set altitude thresholds* …
   Re-write this, e.g. as follows: … longer ray segments reach below the altitude thresholds…

   Done

7. Page 8, lines 10–15: *To account for the relative orientation, we use the quaternions, provided along with the orbits data. … To account for the satellite orientation and maneuvers we use the quaternions, that precisely define the orientation of the satellite with respect to the center of the Earth Centered reference frame at any given time.*
   Quaternions are briefly mentioned here and never more arise in the paper. Quaternions provide one of different representations of the spatial orientation, and they are hardly a unique means for solving the problem in question. They are, therefore, either unnecessary to mention or deserve a deeper explanation.

   Changed

8. Page 9, line 4: *The positive values of $\phi_A$ are defined such as the angle increases towards the positive Y.* Clarify the sentence, e.g. like this: Angle $\phi_A$ has the same sign as Y.

   Done

9. Page 17, line 9–11: *Therefore, instead of a simple average, here we perform a 1 second weighted average where the weight is represented by the SNR value, so that values of $\Delta\Phi$ associated to higher SNR contribute more than those associated to lower SNR.*
   Because SNR and $\Delta\phi$ are correlated random processes, should not this result in a systematic error?

   The value of the SNR could be also understood as a measure proportional to the uncertainty of $\Delta\phi$, so that the higher the SNR, the lower the $\Delta\phi$ uncertainty. But the value of SNR should not affect the $\Delta\phi$ measurement (e.g. constistently high values of SNR are not biasing $\Delta\phi$, but $\Delta\phi$ is more precise). Since those with larger values of SNR should be representing the true measurement with less uncertainty, they weight more in the final average.

[revised manuscript text omitted]

became more evident. During CL tracking that occurred above $\sim$7-9 km altitude, the phase is obtained with half-cycle ambiguity (Ao et al., 2003). During OL tracking that occurred below $\sim$7-9 km altitude, the tracking data are processed on the ground with the 50 Hz navigation modulation removed that enables full-cycle phase reconstruction (Ao et al., 2009; Sokolovskiy et al., 2006) . Therefore, we correct for half-cycle slips during CL and full-cycle slips during OL, as follows: for the CL region, we apply:

$$\Delta\Phi(t) = \arctan(\tan(\Delta\Phi(t))), \tag{2}$$

 and for the OL region we apply:

$$\Delta\Phi(t) = \arctan 2(\sin(\Delta\Phi(t)), (\cos(\Delta\Phi(t))). \tag{3}$$

This approach corrects the half and full cycle slips remaining in the data.  An example of the remaining cycle slips and its correction can be seen in Figure 2.

For each port, data are processed to obtain $N(h)$. To assign a height to each time measurement (e.g. excess phase or SNR) is complicated,  especially when atmospheric multipath is present at the lower layers. To do so we proceed as follows: using geometric optics (GO) retrievals (e.g. Hajj et al., 2002), we obtain an approximated relationship between the impact parameter and received time ($a_{GO}(t)$). This is then used to map the canonical-transformed impact parameter $a_{CT}$ to a unique time $t$. We then rely on the inverse Abel transform and we assign a tangent height (the height of the tangent point of each ray) to each phase and SNR measurement, $\Delta\Phi(h_{\mathrm{t}})$ and $\mathrm{SNR}(h_{\mathrm{t}})$. As we said, this is not exact since the relation $a_{GO}(t)$ contains errors under atmospheric multipath conditions. We have estimated (not shown here) that the uncertainty due to atmospheric multipath varies from 0.1 km at 6 km altitude to 0.6 km at 2 km altitude in the tropics, improving at higher latitudes. Therefore, measurements linked to altitudes lower than 2 km have to be treated with caution.

[revised manuscript text omitted]

10   introduced in our measured $\Delta\Phi$ with respect to $\Delta\Phi_{\text{precip}}$. With the values shown in Fig. 7, we obtain that the measured $\Delta\Phi$ is reduced by a 6% in the case of a $\Omega_2 \sim 10\,\text{deg}$, which would be an extreme and rare situation. In the event that $\Omega_2 = 20\,\text{deg}$, the measured $\Delta\Phi$ would be reduced a 25% below the actual $\Delta\Phi_{\text{precip}}$. The 90% of the observed $\Omega_2$ is confined between -3 and 4.7 deg, which implies that the measured $\Delta\Phi$ is reduced by  1.3 %. It is true that the ionospheric activity has been very low during the period the data was obtained (i.e. 2018-2019), therefore further analyses will be needed when solar

15   activity increases.

[revised manuscript text omitted]

These results confirm the potential of the PRO technique to provide joint measurements of precipitation and thermodynamics, becoming a very valuable and unique technique. Further analyses need to be done in order to address the quantification of precipitation, as well as to exploit this and other scientific applications.

*Competing interests.* The authors declare that they have no conflict of interest.

*Acknowledgements.* R. Padullés research was supported by an appointment to the NASA Postdoctoral Program at the Jet Propulsion Laboratory, administered by Universities Space Research Association under contract with NASA. The JPL co-authors acknowledge support from the NASA US Participating Investigator (USPI) program. The work conducted at ICE-CSIC/IEEC was supported by the Spanish grant ESP2015-70014-C2-2-R. Part of Cardellach's contribution has been supported by the Radio Occultation Meteorology Satellite Application

Facility (ROM SAF) which is a decentralized operational RO processing centre under EUMETSAT. The authors want to thank the two anonymous reviewers for their valuable comments that helped to improve the paper.

---

## Author Comment (AC2) · 3 Feb 2020

**Response to Reviewer 2**

First of all we want to thank the time spent by the reviewers in providing comments and corrections to this manuscript, that are going to greatly improve its quality. Comments from the reviewers made us notice some mistakes in the processing of the data, that have been corrected. Therefore, the data processing has been repeated and more data have been included. All figures have been replaced with the new versions, and even though some might seem very similar, they all now include the whole dataset. The main results have not changed. Few numbers have been changed, with no implications for the conclusions.

Below we specifically address the reviewer's comments one by one. Please note that the text in blue is the original reviewer's comment and in black is our answer.

The manuscript presents the calibration and validation studies for the Radio Occultations and Heavy Precipitation experiment aboard the PAZ satellite. I find the study interesting and well presented. I only have a few minor points:

1. Abstract: please write in words what the 'dphi observable' is

   Done

2. *Page 4, L1: "The JPL designed IGOR+ receiver installed in PAZ collects RO data at a rate of 50Hz. Each RO is tracked independently in the two ports dedicated to the H and V polarized antennas. Therefore, each port output is processed independently. Raw phase data ..."* please write Integrated GPS Occultation Receiver (IGOR) and provide some more details. Tracking modes, setting and rising occultations, etc.?

   Done. We have included more information in Section 2.1

3. Page 4, L27: *"For each port, data are processed to obtain N(h). To assign a height to each time measurement (e.g. excess phase or SNR) is complicated, specially when atmospheric multipath is present. To do so we rely on the inverse Abel transform and we assign a tangent height (the height of the tangent point of each ray) to each phase and SNR measurement, (ht) and SNR(ht) . As a convention, the height that is assigned to each time is the mean of the heights obtained in the H and V ports at that time..."* Are the heights obtained from the H and V ports very different? Are the BA (N) profiles very different? Can you provide an example (maybe in Fig.2 to the right).

   We have included more information on the time to height determination in Section 2.1. We also provide an estimation of the uncertainty due to the atmospheric multipath, that dominates over the possible differences between the H and V derived heights. We prefer not to include examples about the bending angle, because it requires a dedicated calibration different from what is shown here, and is still work in progress.

4. Page 5, L3: *"The whole processing is applied to 59,704 occultations, of which a total of 42,209 pass through the JPL quality control. The quality control is passed if the retrieved refractivity profiles between 0 and 30 km (for both H and V) are within 10% of the co-located NCEP Global Forecast System (GFS). Those that do not pass the quality control are discarded."* I

understand that the focus of this paper is not N profiles, but maybe you could add some N statistics here or later. Also see comment regarding Fig.10 below

See answer to question 8 below.

5. Page 5, L10: *"The ray tracing uses the actual retrieved refractivity profile to account for the bending of the rays."* I suggest to remove the word 'actual'.

   Done

6. Page 13, L8: *"We take two heights, 10 and 50 km, at which we evaluate the Faraday rotation using the co-located values of ne and B from IRI and IGRF"* I suggest to add in brackets the corresponding year. whenever the ionosphere comes into play at least the year (2018/2019) should be mentioned. Maybe you could also mention here or later if you expect some impact during high solar activity

   We have included a reminder that the data was obtained during 2018-2019. It is true that this is a period of low solar activity, and we have mentioned this in the paper. Further analyses will be needed during the periods of high solar anctivity. However, based on the simulations that we performed in Tomàs et al. 2018, even in periods of maximum solar activity we expect the values of Ω to be small enough not be a concern.

7. Page 14, L12: *"...which implies that the measured is reduced by a 1.34 %."* remove 'a' here and i suggest to write 1.3% instead of 1.34%

   Done

8. Page 18, Fig10.: I am curious to see some N statistics. In a recent study Padullés, R., Cardellach, E., Wang, K.-N., Ao, C. O., Turk, F. J., and de la Torre Juárez, M.: Assessment of global navigation satellite system (GNSS) radio occultation refractivity under heavy precipitation, Atmos. Chem. Phys., 18, 11697–11708, https://doi.org/10.5194/acp-18-11697-2018, 2018.:
   *'...This is the aim of polarimetric radio occultations, which will provide joint products of temperature, pressure, and moisture and an indication of the amount of precipitation (mostly sensitive to the heaviest) at each vertical level (Cardellach et al., 2017) with the objective of advancing the understanding of heavy precipitation events, closely linked with high specific-humidities conditions...'*
   Would it be too much of an effort to add to the right the corresponding N statistics (w.r.t. say the NCEP GFS) ? I would like to see if the positive mean deviation correlates with the positive N bias.

   This is certainly a study that has to be done, however, we believe that is out of the scope of this manuscript, which is more focused on the technical aspects of the data. We have performed some very preliminary comparisons like the ones in Padullés et al 2018 and suggested by the reviewer, and we observe more features than what could be easily included here. We believe that including these analyses would deviate too much from the objective of this paper, and to do it right, it would increase the lenght of the paper too much.

We believe a study of this kind could have a dedicated paper with a more thorogh discussion about the science that can be obtained with polarimetric radio occultations.

9. Page 19, L1: *"as we can see in Table 1, 81% of the cases exceed"* write 1.8% instead of 1.81%.

It is a separated word: Table 1, and 81% of cases.

10. Conclusion, L31: *"It is important here to emphasize the fact that we are evaluating the performance in detecting rain rather than quantifying its rate, and the validation in the context of this paper confirms that capability"* but this can be also done with the IMERG product? Can you write something about the potential benefit of your product here or in the introduction.

The potential benefit of polarimetric radio occultations resides in the capability to provide joint measurements of thermodynamics and preciptitaion. Further work on the quantification of preciptiation is needed, so that more scientific applications can be exploited. So far, in this paper, we have confirmed that the detection of precipitation is successful, that the noise level is in agreement with the predictions, and that non-precipitation related effects are small. We have included a last sentence in the conclusions.

[revised manuscript text omitted]

became more evident. During CL tracking that occurred above $\sim$7-9 km altitude, the phase is obtained with half-cycle ambiguity (Ao et al., 2003). During OL tracking that occurred below $\sim$7-9 km altitude, the tracking data are processed on the ground with the 50 Hz navigation modulation removed that enables full-cycle phase reconstruction (Ao et al., 2009; Sokolovskiy et al., 2006) . Therefore, we correct for half-cycle slips during CL and full-cycle slips during OL, as follows: for the CL region, we apply:

$$\Delta\Phi(t) = \arctan(\tan(\Delta\Phi(t))), \tag{2}$$

 and for the OL region we apply:

$$\Delta\Phi(t) = \arctan 2(\sin(\Delta\Phi(t)), (\cos(\Delta\Phi(t))). \tag{3}$$

This approach corrects the half and full cycle slips remaining in the data. An example of the remaining cycle slips and its correction can be seen in Figure 2.

5    For each port, data are processed to obtain $N(h)$. To assign a height to each time measurement (e.g. excess phase or SNR) is complicated,  especially when atmospheric multipath is present at the lower layers. To do so we proceed as follows: using geometric optics (GO) retrievals (e.g. Hajj et al., 2002), we obtain an approximated relationship between the impact parameter and received time ($a_{GO}(t)$). This is then used to map the canonical-transformed impact parameter $a_{CT}$ to a unique time $t$. We then rely on the inverse Abel transform and we assign a tangent height (the height of the tangent point of each ray)

10   to each phase and SNR measurement, $\Delta\Phi(h_t)$ and $\text{SNR}(h_t)$. As we said, this is not exact since the relation $a_{GO}(t)$ contains errors under atmospheric multipath conditions. We have estimated (not shown here) that the uncertainty due to atmospheric multipath varies from 0.1 km at 6 km altitude to 0.6 km at 2 km altitude in the tropics, improving at higher latitudes. Therefore, measurements linked to altitudes lower than 2 km have to be treated with caution.

As a convention, the height that is assigned to each time is the mean of the heights obtained in the H and V ports at that

15   time. To set a common reference for all the data that is independent on the initial phase of the receiver, we set the zero at 30 km, and therefore $\Delta\Phi = \Delta\Phi - \Delta\Phi(h_t = 30)$. At this height we know that there is no rain, clouds or ice that could infer any measurable differential phase shift. Therefore, all measurements are relative to that height.

The whole processing is applied to  96,446 occultations collected between 2018-05-10 and 2019-10-10, of which a total of  74,604 pass through the JPL quality control. The quality control is passed if the retrieved refractivity

20   profiles between 0 and 30 km (for both H and V) are within 10% of the co-located NCEP Global Forecast System (GFS). Those that do not pass the quality control are discarded.

**2.2   Ray tracing**

In order to identify the region that is being sensed by the PRO, we need to define the RO plane. The RO plane is formed by all the rays from the GPS transmitter to the receiver. This plane is slant rather than vertical due to the relative movement between

25   the GPS and the LEO, which are not coplanar. To define a realistic RO observation plane, we account for realistic rays between the GPS and the LEO obtained using a ray tracing software that provides the ray's trajectories for every time step of the PRO event. The ray tracing uses the  retrieved refractivity profile to account for the bending of the rays.

The whole set of trajectories, e.g. (time, lon, lat, height), can be used to identify the regions traversed by the rays, and therefore perform realistic and accurate co-locations between different datasets (like precipitation) for reliable calibration and

30   validation of the experiment.

**2.3   Co-location of PRO observations with GPM constellation products**

For the calibration and validation part of the experiment, the co-locations with precipitation products is crucial. It provides an independent measure on whether an observation might have been affected by rain or not. Since the effect of rain is the objective

[Figure]

**Figure 2.** Example of one Polarimetric RO observation, corresponding to id 20180827_2108paz_gps57. The RO tangent point is located at (46.5N, 165.3E). (Top) SNR for H (black) and V (red) ports as a function of time. (Middle) Raw differential phase shift between the H and V excess phase observables (black), and the same observable after being corrected for cycle slips, following the procedure in Sect. 2.1. Notice that before the CL to OL transition (gray vertical line), there are two half cycles slips (jumps of $\pi$), while after the transition, several full cycle slips (jumps of $2\pi$) appear. (Bottom) Corrected differential phase shift (
[revised manuscript text omitted]

Such expressions are thoroughly derived in Tomás et al. (2018). It is also shown in Tomás et al. (2018), based on simulations, that the effect of the Faraday Rotation due to the impurities in the emission (Eq. 9) should be possible to correct, and the effect

5    after crossing rain (Eq. 10) is small enough to not introduce substantial errors in the measurements because $\Omega_2$ is generally low.

In the following section we analyze the observations based on the co-located electron density and magnetic field in order to infer whether the ionosphere is inducing any noticeable $\Delta\Phi$.

**4.1 Faraday Rotation for PAZ PRO events**

10    First of all we need to know what are the typical values of Faraday Rotation along PRO rays. We take two heights, 10 and 50 km, at which we evaluate the Faraday rotation using the co-located values of $n_e$ and $B$ from IRI and IGRF. For every PRO, we compute the total Faraday Rotation at 50 and 10 km, and the second part of the Faraday Rotation at 10 km. The histogram for all the cases is shown in Fig. 7

Summarizing Fig. 7, the  the total Faraday Rotation has values between -12 and 20 deg, and between -6

15    and 10 for the portion of the Faraday Rotation between the tangent point and the receiver. We have also seen that the maximum

[revised manuscript text omitted]

**5.1 Smoothing of the signal**

Once we have calibrated the signal, we smooth it to reduce the uncertainty. PRO are acquired at 50 Hz, but for the purposes of detecting precipitation it is enough to have measurements at 1 s resolution. While the smoothing reduces the standard error by accounting for more samples for each measurement (e.g. we use 50 points obtained at 50 Hz to represent the measurement at

$(1) \quad \Delta\Phi(t) = \phi_H(t) - \phi_V(t)$

$t \to \theta_A, \varphi_A, h \quad (2)$

$(3)$
$\mathrm{CL}: \Delta\Phi(t) = \arctan(\tan(\Delta\Phi(t)))$
$\mathrm{OL}: \Delta\Phi(t) = \arctan2(\sin(\Delta\Phi(t)), \cos(\Delta\Phi(t)))$

$\Delta\Phi^{\mathrm{rain}}$
$\Delta\Phi^{\mathrm{no-rain}} \quad (5)$

$(4) \quad \Delta\Phi(h) = \Delta\Phi(h) - \Delta\Phi(h = 30km)$

$\Omega; \ \Omega_2 \quad (6)$

$(7) \quad \langle\Delta\Phi^{\mathrm{no-rain}}_{\mathrm{low-iono}}(\theta_A, \varphi_A)\rangle_{\mathrm{pattern}}$

$(8) \quad \Delta\Phi^i_c = \Delta\Phi^i(\theta_A, \varphi_A) - \langle\Delta\Phi^{\mathrm{no-rain}}_{\mathrm{low-iono}}(\theta_A, \varphi_A)\rangle_{\mathrm{pattern}}$

$(9) \quad \Delta\Phi^i = \Delta\Phi^i_c(h) - \mathrm{trend}(\Delta\Phi^i_c(h > 20km))$

**Figure 9.** Block diagram identifying and describing the steps followed during calibration: (1) observable: phase difference between H and V; (2)  mapping time into other variables; (3) correction for remaining cycle slips; (4) set the zero level at 30 km; (5) and (6) co-locations with precipitation information and ionospheric activity; (7) accumulation of free of rain and low ionospheric activity measurements to create the effective antenna pattern; (8) subtraction of the effective antenna pattern to each measurement; (9) correcting for remaining trends.

1 s resolution), its counterpart is that it reduces the vertical resolution of the observation, being of about few hundred meters after smoothing. Generally, a simple running average window would be applied to perform the smoothing. However, here we want to stress the fact that the measurements with higher Signal to Noise Ratio have less uncertainty. The uncertainty in the phase measurement is determined by the SNR of each measurement (e.g. Cardellach et al., 2014), so the uncertainty of the $\Delta\Phi$ comes from the propagation of such error from both H and V ports. Therefore, instead of a simple average, here we perform a 1 second weighted average where the weight is represented by the SNR value, so that values of $\Delta\Phi$ associated to higher SNR contribute more than those associated to lower SNR. In this case, since we are combining both the measurements from the H and V ports, the SNR that we use for the weighted average is: $\mathrm{SNR} = (\mathrm{SNR}_H + \mathrm{SNR}_V)/\sqrt{2}$. The SNR values are limited so that only those above 10 V/V contribute to the mean.

**6 Validation of the $\Delta\Phi$**

The smoothed calibrated measurements $\langle\Delta\Phi\rangle_{1s}$ are to be validated against IMERG. For comparison and standardization purposes, we interpolate the $\langle\Delta\Phi\rangle_{1s}$ into a 100 m grid spacing profiles from 0 to 30 km: $\Delta\Phi(\mathrm{h}_{100m})$. For these profiles we can perform the mean and the standard deviation at each altitude. First of all, we group them by their linked precipitation and brightness temperature: (1) no-precipitation ($R = 0\,mm/h$ and $Tb > 250\,K$), precipitation ($R > 0.1\,mm/h$), and heavy precipitation ($R > 1\,mm/h$). For these three groups we compute the mean and standard deviation as a function of height. In Fig. 10 we show the results. We can see how for the no-precipitation group, the $\Delta\Phi(\mathrm{h}_{100m})$ averages to 0 for the whole vertical profile (by design), and the standard deviation ($\sigma_{\Delta\Phi}(h)$) increases with decreasing altitudes. In the right panel of Fig. 10 we show a more detailed profile of the $\sigma_{\Delta\Phi}(h)$.

The first remark is that  $\sigma_{\Delta\Phi}(h) = 1.2\,\mathrm{mm}$ at 2 km of height. This is better than the study performed in Cardellach et al. (2019), expected since here we calibrated the signal using the antenna pattern and we have performed the weighted average smoothing. It is also very close to the theoretically predicted sensitivity in Cardellach et al. (2014). The second noticeable feature is the peak in $\sigma_{\Delta\Phi}(h)$ around 7 km. This feature is due to instrumental effects in some occultations near the  CL to OL transition, and is an open issue under investigation. Finally, a small negative bias is observed within the lower 1 km. As we have explained in Sect. 2.1, the results below 2 km have to be treated with caution, since many uncertainties (tangent height determination, atmospheric multipath, etc.) come into play.

[revised manuscript text omitted]

These results confirm the potential of the PRO technique to provide joint measurements of precipitation and thermodynamics, becoming a very valuable and unique technique. Further analyses need to be done in order to address the quantification of

25   precipitation, as well as to exploit this and other scientific applications.

*Competing interests.* The authors declare that they have no conflict of interest.

*Acknowledgements.* R. Padullés research was supported by an appointment to the NASA Postdoctoral Program at the Jet Propulsion Laboratory, administered by Universities Space Research Association under contract with NASA. The JPL co-authors acknowledge support from the NASA US Participating Investigator (USPI) program. The work conducted at ICE-CSIC/IEEC was supported by the Spanish grant

30   ESP2015-70014-C2-2-R. Part of Cardellach's contribution has been supported by the Radio Occultation Meteorology Satellite Application

Facility (ROM SAF) which is a decentralized operational RO processing centre under EUMETSAT. The authors want to thank the two anonymous reviewers for their valuable comments that helped to improve the paper.